# DuRP: Dual-Stage Physics-Embedded Learning for Joint Radiance and Polarization Restoration

**Zhenshuo Yang** [1,2]  **Qian He** [1,2]  **Zhiyuan Liu** [1,2]  **Baojie Fan** [3]  **Jiandong Tian** [1]

## Abstract

Polarization information is valuable for many computer vision applications. However, in hazy environments, polarization information is severely attenuated due to the degradation of captured polarized images. Existing dehazing methods struggle to effectively restore polarization information, as single-image methods are unaware of polarization, and polarization-based methods are constrained by the traditional polarization models. These deficiencies lead to inaccurate polarimetric signatures and physical inconsistencies in scattering environments. To overcome these limitations, we propose DuRP, a dual-stage physics-embedded learning framework for joint restoration of scene radiance and polarization information. Specifically, we derive generalized polarization physics models that relax the ideal assumptions of traditional theory to provide a more precise foundation for the joint restoration of polarimetric and amplitude information. We then design a dual-stage neural network to estimate latent physical parameters through differentiable operators, ensuring that both the polarimetric state and radiance are accurately recovered. Experimental results show that DuRP achieves state-of-the-art performance in joint restoration and significantly enhances polarization-based vision tasks. Project website: https://DuRP.github.io/.

## 1. Introduction

Polarization information provides critical physical cues for various computer vision tasks, such as material segmentation (Liang et al., 2022), vehicle detection (Dong et al.,

[1]State Key Laboratory of Robotics and Intelligent Systems, Shenyang Institute of Automation, Chinese Academy of Sciences. [2]University of the Chinese Academy of Sciences. [3]Nanjing University of Posts and Telecommunications. Correspondence to: Jiandong Tian <tianjd@sia.cn>, Baojie Fan <jobfbj@gmail.com>.

*Proceedings of the 43rd International Conference on Machine Learning*, Seoul, South Korea. PMLR 306, 2026. Copyright 2026 by the author(s).

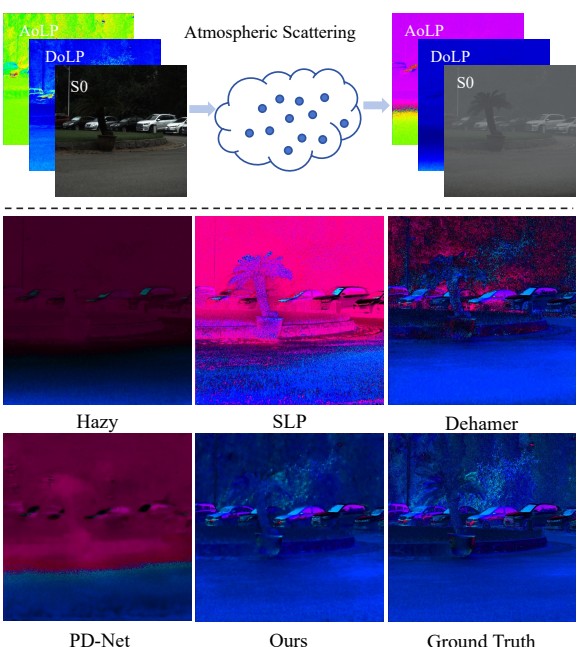

*Figure 1.* Joint restoration in scattering media. Top. Scattering degrades radiance and polarization. Bottom. *DuRP* recovers the radiance and polarimetric state via generalized physics models. Notably, DoLP and AoLP are composited into a single image.

2024), industrial film removal (Tang et al., 2024), and 3D reconstruction (Deschaintre et al., 2021; Lyu et al., 2024). However, in scattering media like haze, images captured by widely used Division of Focal Plane (DoFP) polarimetric cameras suffer severe degradation. This atmospheric scattering distorts both radiance and polarization signals, including the Degree of Linear Polarization (DoLP) and Angle of Linear Polarization (AoLP), as shown in Figure. 1 (top). Accurate recovery of both polarization and radiance is therefore essential to improve the reliability of polarization-based vision systems in complex environments.

Recent advances in single-image dehazing (He et al., 2011; Berman et al., 2016; Guo et al., 2022; Ling et al., 2023; Chen et al., 2024b; Feng et al., 2024; Wang et al., 2024b; Chen et al., 2024a) have achieved impressive results in radiance restoration. However, when applied to polarized hazy images, these methods treat each polarization channel independently and neglect the intrinsic physical constraints

among them, which leads to inaccurate and inconsistent polarization reconstructions.

In contrast, polarization-based dehazing methods possess the inherent capacity to jointly process multiple polarized inputs, yet they remain hindered by fundamental obstacles in recovering accurate scene polarization. A primary limitation stems from the prevailing focus on leveraging polarimetric cues for radiance enhancement (Shen et al., 2018; Gao et al., 2023; Ma et al., 2024; Li et al., 2024b; Huang et al., 2024) , which often leaves the explicit reconstruction of the polarization state itself overlooked. This challenge is further exacerbated by the widespread reliance on traditional models (Treibitz & Schechner, 2009; Fang et al., 2014; Zhou et al., 2021; Yang et al., 2025) that assume an invariant AoLP across both the scene and the atmosphere. Such restrictive assumptions render the recovery of true scene AoLP theoretically impossible; moreover, due to the intrinsic coupling between AoLP and DoLP, they inevitably propagate significant errors into both DoLP and radiance estimation (as discussed in Section 3.2).

To overcome these limitations, we propose **DuRP**, a **Dual-Stage Physics-Embedded Learning** framework that explicitly incorporates generalized polarization physics as differentiable inductive biases. By re-deriving the polarimetric scattering imaging process, we formulate two generalized physical models, namely the Polarization Reconstruction Model and the Radiance Reconstruction Model. Unlike traditional methods, our models relax the assumption of invariant AoLP, providing a more precise foundation for the joint restoration of polarimetric and radiance information. Building upon these physical models, we design a novel two-stage neural network architecture that learns to estimate latent physical parameters in an end-to-end manner without requiring handcrafted or image-specific priors. Each stage performs an initial reconstruction guided by explicit physical constraints, which is further refined through learned feature representations to ensure both numerical accuracy and physical consistency. Consequently, this process enables a more robust recovery of the scene's intrinsic polarization and radiance, as demonstrated in Figure 1 (bottom).

Existing datasets (Zhou et al., 2021) often rely on over-simplified assumptions, such as assigning uniform pseudo-polarization to semantically similar objects. This contradicts real-world DoLP variations (Fig. 2(a)) and fails to capture the complex physical correlations required by our generalized models. To bridge this gap, we propose a novel synthesis pipeline (Fig. 2(b)) and construct the SRPG dataset to provide a realistic training foundation for DuRP.

Extensive experiments demonstrate that DuRP achieves state-of-the-art (SOTA) performance in both polarization restoration and downstream vision tasks. Our primary contributions are:

- We propose DuRP, a physics-embedded framework that integrates generalized polarimetric models as differentiable inductive biases, enabling physically consistent signal restoration.

- We derive generalized polarization physics models that relax the restrictive assumption of invariant AoLP, providing a mathematically rigorous foundation for joint radiance and polarization recovery.

- We construct the SRPG dataset via a novel synthesis pipeline that realistically simulates complex polarimetric distributions, bridging the gap between simplified simulations and authentic scattering environments.

## 2. Related Work

**Single Image Dehazing.** Classical dehazing approaches (He et al., 2011; Berman et al., 2016; Ling et al., 2023; Li & Zhang, 2024) rely on handcrafted physical priors to invert the scattering process. The advent of deep learning has shifted this field toward data-driven representation learning, reflecting a broader trend of learning structured models of visual environments (Dong et al., 2026). Representative methods include data-driven models (Cai et al., 2016; Li et al., 2017; Zhang & Patel, 2018; Chen et al., 2020; Guo et al., 2022) and end-to-end architectures (Song et al., 2023; Li et al., 2023; Chen et al., 2024b; Feng et al., 2024; Chen et al., 2024a) that primarily focus on radiance restoration. While effective for haze removal, these radiance-oriented paradigms remain insufficient for recovering polarimetric information. This limitation arises from a fundamental lack of polarimetric inductive biases; specifically, treating polarization channels independently neglects the intrinsic inter-channel constraints for consistent polarimetric recovery.

**Polarization Dehazing.** Polarization-based methods (Schechner et al., 2001; Treibitz & Schechner, 2009; Miyazaki et al., 2013; Fang et al., 2014; Liang et al., 2015; Huang et al., 2016; Shen et al., 2018; Liang et al., 2020; Gao et al., 2023; Ma et al., 2024; Li et al., 2024b; Huang et al., 2024) leverage the inherent properties of scattered light to enhance scene visibility. Despite their effectiveness in radiance recovery, these approaches predominantly treat polarization as an auxiliary cue rather than a primary restoration target. Furthermore, existing methods (Treibitz & Schechner, 2009; Fang et al., 2014; Zhou et al., 2021; Yang et al., 2025) are often bottlenecked by a model mis-specification—the restrictive assumption that the AoLP of airlight is identical to that of the scene light. This idealization renders the recovery of true scene AoLP theoretically impossible and, due to the non-linear coupling between AoLP and DoLP, induces significant error propagation across the estimated signals. Additionally, current public polarized haze datasets (Zhou et al., 2021) rely on

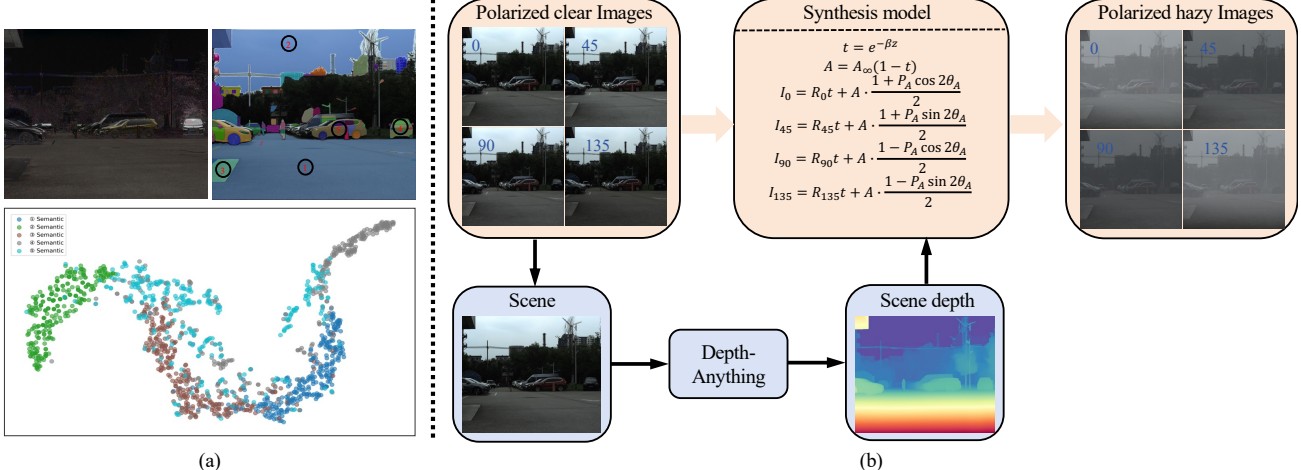

*Figure 2.* Motivation and pipeline for the SRPG dataset. (a) t-SNE visualization highlights significant DoLP diversity among objects within identical semantic categories. (b) Overview of the proposed synthesis pipeline grounded in generalized scattering physics.

pseudo-polarization labels that fail to capture the complex physical distributions observed in real-world scattering scenarios, leading to generalization failure. Instead, our work treats the restoration as a joint inference problem within a general physical framework, ensuring consistent recovery of both radiance and polarization.

## 3. Methodology

### 3.1. Generalized Polarimetric Scattering Model

In scattering media, image quality is significantly degraded by the interactions between light and suspended particles. This process is typically governed by the atmospheric scattering model (Nayar & Narasimhan, 1999) as follows:

$$I(\mathbf{u}) = D(\mathbf{u}) + A(\mathbf{u}), \tag{1}$$

$$D(\mathbf{u}) = R(\mathbf{u})t(\mathbf{u}), \tag{2}$$

$$A(\mathbf{u}) = A_\infty(1 - t(\mathbf{u})), \tag{3}$$

where $\mathbf{u} = (x, y)$ represents the pixel coordinates, $I$ is the observed intensity, $D$ is the attenuated scene radiance, $A$ is the airlight, $R$ is the haze-free radiance, $t$ is the transmission map, and $A_\infty$ is the atmospheric light. For a DoFP polarimetric camera, the intensity measured at analyzer angle $\alpha \in \{0°, 45°, 90°, 135°\}$ in haze is given by:

$$I_\alpha(\mathbf{u}) = I_u(\mathbf{u}) + I_p(\mathbf{u}), \tag{4}$$

where $I_u$ represents the unpolarized light, and $I_p$ denotes the linearly polarized light. According to Malus's Law, these two components can be further expressed as:

$$I_u(\mathbf{u}) = \frac{1}{2}I(\mathbf{u})(1 - P(\mathbf{u})), \tag{5}$$

$$I_p = I(\mathbf{u})P(\mathbf{u})\cos^2(\alpha - \theta(\mathbf{u})), \tag{6}$$

where $P$ represents the DoLP, and $\theta$ represents the AoLP. Substituting Eqs. (5)(6) into Eq. (4) yields the general response of a DoFP sensor to partially linearly polarized light:

$$I_\alpha(\mathbf{u}) = \frac{I(\mathbf{u})\{1 + P(\mathbf{u})\cos[2(\alpha - \theta(\mathbf{u}))]\}}{2}. \tag{7}$$

Combining the Eq. (7) with the atmospheric scattering model (Eqs. (1)-(3)), the joint polarization-dependent degradation at $\alpha \in \{0°, 45°, 90°, 135°\}$ can be formulated as:

$$
\begin{aligned}
I_0(\mathbf{u}) &= R_0(\mathbf{u})t(\mathbf{u}) + A(\mathbf{u})\frac{1 + P_A(\mathbf{u})\cos 2\theta_A}{2}, \\
I_{45}(\mathbf{u}) &= R_{45}(\mathbf{u})t(\mathbf{u}) + A(\mathbf{u})\frac{1 + P_A(\mathbf{u})\sin 2\theta_A}{2}, \\
I_{90}(\mathbf{u}) &= R_{90}(\mathbf{u})t(\mathbf{u}) + A(\mathbf{u})\frac{1 - P_A(\mathbf{u})\cos 2\theta_A}{2}, \\
I_{135}(\mathbf{u}) &= R_{135}(\mathbf{u})t(\mathbf{u}) + A(\mathbf{u})\frac{1 - P_A(\mathbf{u})\sin 2\theta_A}{2},
\end{aligned}
\tag{8}
$$

where $I_\alpha$ and $R_\alpha$ denote the degraded and latent intensities at $\alpha \in \{0°, 45°, 90°, 135°\}$, while $(P_A, \theta_A)$ defines the airlight's polarimetric signature. This generalized formulation provides the physical foundation for our synthesis pipeline (Figure. 2(b)) and subsequent reconstruction.

### 3.2. Polarimetric Phasor Formulation

To analyze the polarimetric degradation, we introduce the Stokes vector to provide a compact representation of light states. For any signal $i \in \{R, D, A, I\}$, its Stokes vector is defined as $\mathbf{S}^i = [S_0^i, S_1^i, S_2^i, S_3^i]^\mathrm{T}$, where the components are related to the polarized intensities by:

$$
\begin{aligned}
S_0^i &= i_0 + i_{90} = i_{45} + i_{135}, \\
S_1^i &= i_0 - i_{90}, \\
S_2^i &= i_{45} - i_{135},
\end{aligned}
\tag{9}
$$

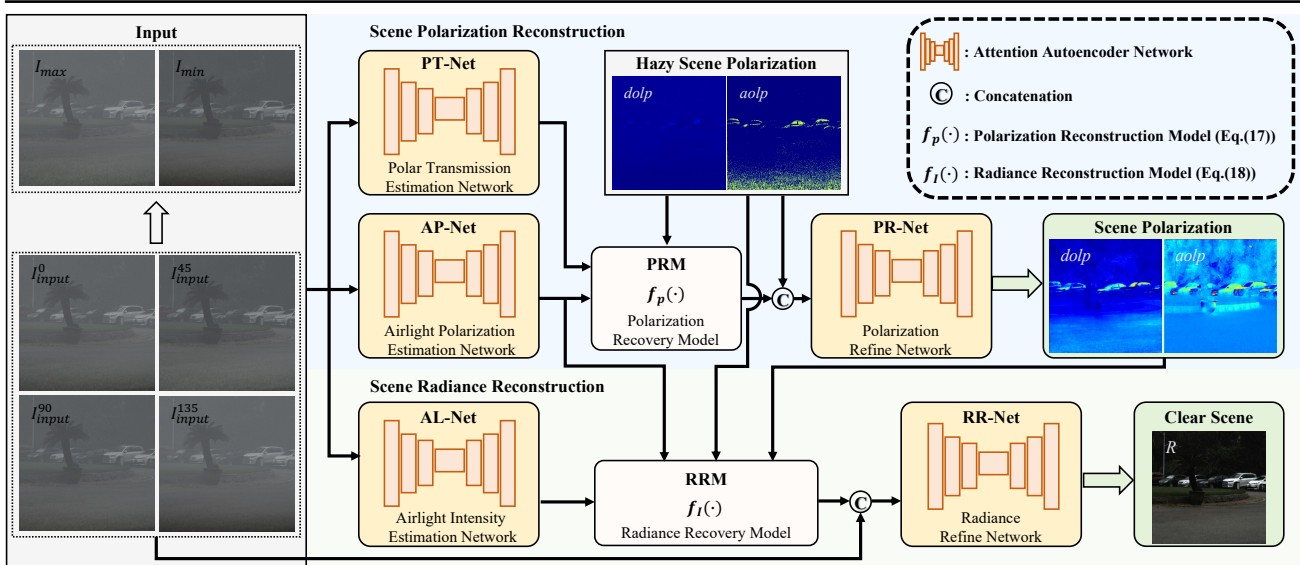

*Figure 3.* Overview of the DuRP framework. The dual-stage architecture sequentially performs Scene Polarization Reconstruction (P-Stage) and Scene Radiance Reconstruction (R-Stage). In the P-Stage, PT-Net and AP-Net estimate the polarization transmission $K$ and airlight $C_A$, which are embedded into the differentiable PRM to recover the scene polarization $C_D$. In the R-Stage, AL-Net and the RRM leverage these outputs to restore the scene radiance $R$. Both stages incorporate refinement modules to ensure optimal signal accuracy and physical consistency.

Since circular polarization $S_3^i$ is negligible in natural outdoor scenes, we set $S_3^i = 0$. Accordingly, the DoLP and AoLP can be computed as:

$$P_i = \frac{\sqrt{(S_1^i)^2 + (S_2^i)^2}}{S_0^i},$$
$$\theta_i = \frac{1}{2}\operatorname{atan2}(S_2^i, S_1^i). \tag{10}$$

By integrating the Eqs. (1)-(3) with the Eq. (8), the Stokes vector of the hazy image, $\mathbf{S}^I$, can be derived via Eq. (9) as:

$$S_0^I = S_0^D + S_0^A = S_0^R t + A,$$
$$S_1^I = S_1^D + S_1^A = S_1^R t + AP_A \cos 2\theta_A, \tag{11}$$
$$S_2^I = S_2^D + S_2^A = S_2^R t + AP_A \sin 2\theta_A,$$

where $\mathbf{S}^R$ is the latent scene Stokes vector and $A = A(\mathbf{u}) = A_\infty(1 - t(\mathbf{u}))$. Combining Eq. (10) and Eq. (11), we can derive the relationship between $P_A$, $P_D$, and $P_I$ as follows:

$$IP_I = \sqrt{(DP_D + AP_A)^2 + 2AP_A DP_D \xi},$$
$$\xi = \cos 2\theta_D \cos 2\theta_A + \sin 2\theta_D \sin 2\theta_A - 1, \tag{12}$$

here $I = S_0^I$, $D = S_0^R t$, $P_D = P_R$, and $\theta_D = \theta_R$. Similarly, we can obtain the relationship between $\theta_A$, $\theta_D$ and $\theta_I$ as follows[1]:

$$\tan 2\theta_I = \frac{DP_D \sin 2\theta_D + AP_A \sin 2\theta_A}{DP_D \cos 2\theta_D + AP_A \cos 2\theta_A}. \tag{13}$$

To unify and simplify these coupled equations, we extend the DoLP and AoLP representation into the complex domain. Eq. (12) and Eq. (13) are equivalent as follows:

$$|IP_I \cdot e^{\mathrm{j}2\theta_I}| = |AP_A e^{\mathrm{j}2\theta_A} + DP_D e^{\mathrm{j}2\theta_D}|$$
$$\arg(IP_I \cdot e^{\mathrm{j}2\theta_I}) = \arg(AP_A e^{\mathrm{j}2\theta_A} + DP_D e^{\mathrm{j}2\theta_D}), \tag{14}$$

which can be equivalently written as[2]:

$$I \cdot P_I e^{\mathrm{j}2\theta_I} = A \cdot P_A e^{\mathrm{j}2\theta_A} + D \cdot P_D e^{\mathrm{j}2\theta_D}. \tag{15}$$

Notably, Eq. (15) generalizes classical polarimetric models (Zhou et al., 2021; Yang et al., 2025) by relaxing the assumption of identical orientations for airlight and scene ($\theta_A = \theta_D$). To ensure computational tractability and capture this generalized physics, we adopt the phasor representation $C_i = P_i e^{\mathrm{j}2\theta_i}$ as:

$$I \cdot C_I = A \cdot C_A + D \cdot C_D. \tag{16}$$

Eq. (16) reformulates the complex scattering-induced coupling as a linear phasor mixture. By mapping periodic AoLP into a continuous complex domain, this re-parameterization enables the implementation of physics as differentiable operators, ensuring stable end-to-end optimization. Based on this linearized physics, we analytically derive the Polarization Reconstruction Model (PRM) and Radiance Reconstruction Model (RRM) as follows:

$$C_D = \frac{(C_I - C_A)}{K} + C_A, \tag{17}$$

---

[1]Please refer to the appendix A for more details.

[2]Please refer to the appendix B for more details.

$$R = \frac{K \cdot I A_\infty}{A_\infty + I \cdot (K-1)}. \tag{18}$$

Here, the polarization transmission $K = D/I$ is constrained to $[\epsilon, 1]$ for numerical stability. As underscored by Eqs. (17)–(18), the non-linear coupling between DoLP and AoLP renders independent channel restoration and restoration based on traditional models insufficient[3], thereby necessitating the joint optimization of latent physical variables to ensure physically consistent results.

### 3.3. Dual-Stage Physics-Embedded Learning

The proposed DuRP framework integrates the analytical image-formation models (Eqs. (17)-(18)) as differentiable operators to jointly reconstruct scene polarization and radiance. Rather than employing a black-box mapping, DuRP decomposes the inverse problem into two interlocking stages: Scene Polarization Reconstruction (P-Stage) followed by Scene Radiance Reconstruction (R-Stage), as shown in Figure. 3. This cascaded design ensures that the physical parameters estimated in the first stage provide a rigorous foundation for the subsequent radiance restoration.

**Network Input.** In addition to the four polarimetric images $I_\alpha$, we concatenate the max and min intensity representations $(I_{\max}, I_{\min})$ to the input stream. Computed as:

$$\begin{aligned} I_{max} &= \frac{S_0^I + \sqrt{(S_1^I)^2 + (S_2^I)^2}}{2}, \\ I_{min} &= \frac{S_0^I - \sqrt{(S_1^I)^2 + (S_2^I)^2}}{2}. \end{aligned} \tag{19}$$

These components provide rotation-invariant features that encode the intrinsic extrema of the polarimetric intensity, ensuring that the feature representation is coordinate-independent[4]. This representation helps the network distinguish between scene-intrinsic polarization and scattering-induced degradation, which has proven effective in various polarimetric vision tasks (Li et al., 2024a; Tang et al., 2024).

**Scene Polarization Reconstruction (P-Stage).** The objective of the P-Stage is to recover the scene polarization $C_D$ by solving Eq. (17). This requires estimating the polarization transmission $K$ and airlight polarization $C_A$ through two specialized sub-networks, PT-Net ($f_{PT}$) and AP-Net ($f_{AP}$):

$$K, C_A = f_{PT}(I_\alpha, I_{\max/\min}), \quad f_{AP}(I_\alpha, I_{\max/\min}). \tag{20}$$

These estimates are embedded into the differentiable PRM operator to yield an initial polarization estimate $\widehat{C_D}$. To compensate for potential non-ideal factors not captured by the physical model (e.g., multi-scattering), a refinement network (SPR-Net) produces the final polarization: $C_D = f_{SPR}(\widehat{C_D}, C_I)$.

---

[3]Please refer to the appendix C.1 for more details.

[4]Please refer to the appendix C.2 for more details.

**Scene Radiance Reconstruction (R-Stage).** Given the restored $C_D$, the R-Stage reconstructs the scene radiance $R$ by estimating the atmospheric light $A_\infty$ via AL-Net ($f_{AL}$). The variables $\{A_\infty, C_A, C_D, I_\alpha\}$ are then fused through the differentiable RRM operator (Eq. (18)) to generate an initial radiance $\widehat{R}$. A final refinement module (SRR-Net) is employed to enhance local texture details and ensure global contrast: $R = f_{SRR}(\widehat{R}, I_\alpha)$.

**Loss and Optimization.** To ensure physical consistency, we define a composite loss $\mathcal{L}(x)$ as a hybrid of $\ell_1$ and $\ell_2$ norms:

$$\mathcal{L}(x) = \|x - \hat{x}\|_1 + 2\|x - \hat{x}\|_2^2. \tag{21}$$

For real-valued quantities (e.g., $K$), the norms apply directly. For complex phasors $x \in \{C_A, C_D\}$, norms are applied independently to the real and imaginary parts rather than to the complex modulus, giving the explicit expansion:

$$\begin{aligned} \mathcal{L}(x) = &\big(\|\mathrm{Re}(x-\hat{x})\|_1 + \|\mathrm{Im}(x-\hat{x})\|_1\big) \\ &+ 2\big(\|\mathrm{Re}(x-\hat{x})\|_2^2 + \|\mathrm{Im}(x-\hat{x})\|_2^2\big). \end{aligned} \tag{22}$$

We employ a three-stage optimization strategy to train the **DuRP** framework. In P-Stage, polarization reconstruction is supervised via:

$$\mathcal{L}_P = \sum_{x \in \{K, C_A, C_D\}} \lambda_x \mathcal{L}(x), \tag{23}$$

where weights $\{\lambda_K, \lambda_{C_A}, \lambda_{C_D}\}$ are empirically set to $\{1, 1, 5\}$. The elevated weight on $C_D$ prioritizes the accuracy of the recovered scene polarization. In R-Stage, the radiance branch is optimized using:

$$\mathcal{L}_R = \mathcal{L}(A_\infty) + 2\,\mathcal{L}(R). \tag{24}$$

This stage balances atmospheric light estimation with radiance restoration. During Joint Fine-tuning, following independent pretraining, the framework is optimized end-to-end:

$$\mathcal{L}_{\text{total}} = \mathcal{L}(C_D) + 1.2\,\mathcal{L}(R). \tag{25}$$

This final stage resolves coupling errors and ensures consistency between the polarization and radiance branches.

## 4. Experiments

### 4.1. Dataset and Baselines

**Dataset.** To bridge the gap in polarimetric dehazing benchmarks, we introduce the SRPG dataset, comprising 387 training and 43 testing real-world scenes. Following the pipeline in Figure. 2, we generate 1,290 polarization image pairs. Specifically, clear outdoor scenes are captured via a FLIR Blackfly BFS-U3-51S5P-C camera. To ensure geometric consistency, we leverage DepthAnything (Yang et al., 2024) to estimate high-fidelity relative depth for transmission map synthesis. The degraded polarization images are

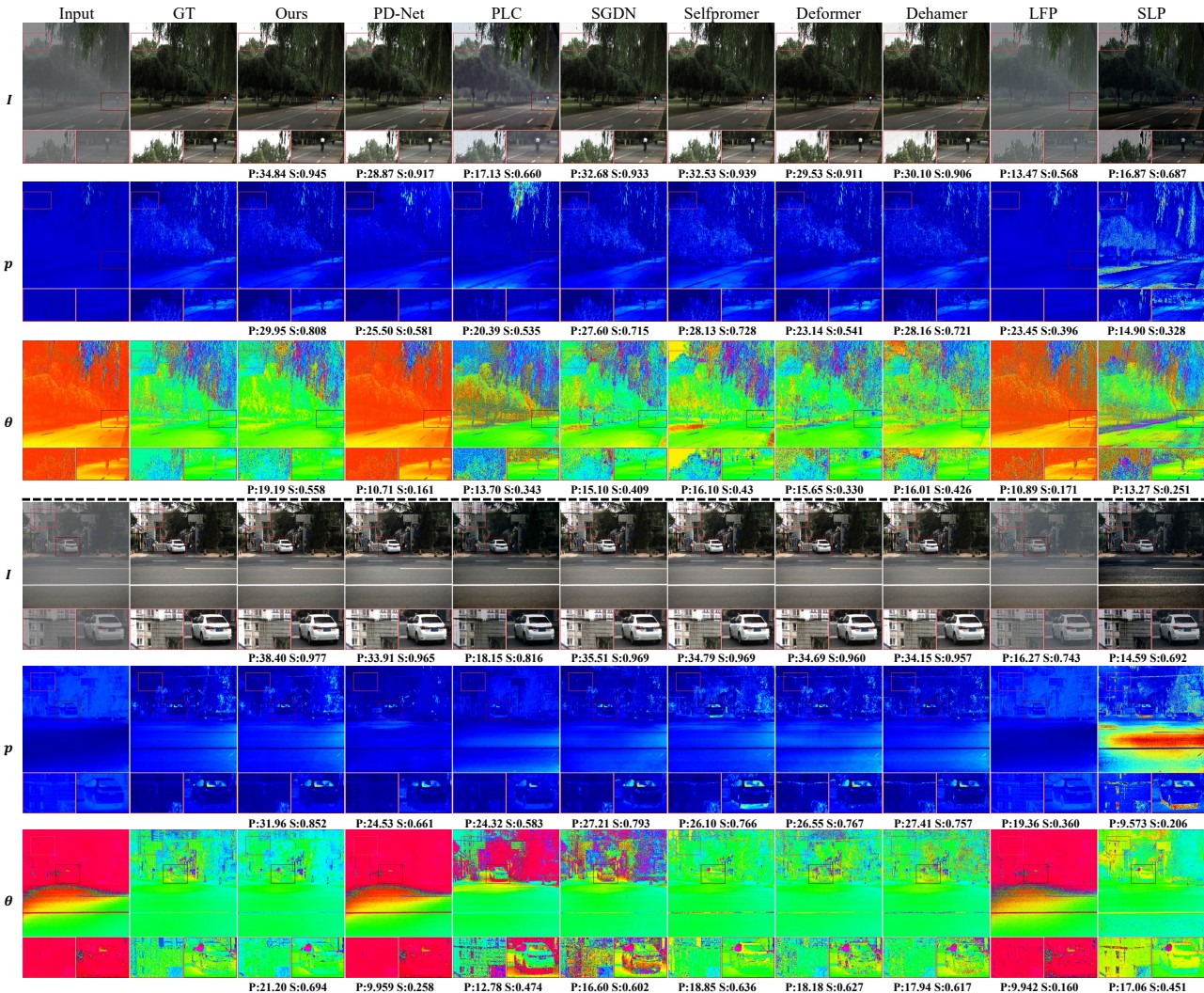

*Figure 4.* Qualitative results on synthetic data. Quantitative metrics (PSNR and SSIM) are provided beneath each instance. For polarimetric signatures, DoLP ($P$) and AoLP ($\theta$) are visualized using normalized colormaps derived from channel-averaged Stokes vectors. *Best viewed by zooming in.*

then generated via our generalized physical model (Eq. (8)), capturing complex real-world scattering dynamics. The detailed generation of SRPG is provided in the appendix D.1

**Baselines.** As no established benchmark exists for polarization restoration, we evaluate DuRP against six state-of-the-art single-image dehazing models: SLP (Ling et al., 2023), LFP (Li & Zhang, 2024), Dehamer (Guo et al., 2022), Deformer (Song et al., 2023), Selfpromer (Wang et al., 2024a), and SGDN (Fang et al., 2025). We also compare two polarization-based methods, PLC (Su et al., 2024) and PD-Net (Zhou et al., 2021), for comparison. To ensure fairness, all learning-based models are retrained on our SRPG dataset. To adapt single-image dehazing models for polarization estimation, each degraded image at a specific polarization angle is independently processed to reconstruct the corresponding clear image at the same angle.

### 4.2. Details in Implementation

The DuRP framework is implemented in PyTorch and trained on an NVIDIA RTX 4090 GPU under Ubuntu 22.04. We employ a three-stage training strategy to ensure convergence: (i) the **P-Stage** and **R-Stage** are trained independently in parallel to initialize the physical parameters; (ii) a joint fine-tuning stage then optimizes the entire network end-to-end. Each stage is trained for 400 epochs using the Adam optimizer ($\beta_1 = 0.9, \beta_2 = 0.999$) with a weight decay of $10^{-4}$. The learning rate begins at $10^{-4}$ ($10^{-5}$ for fine-tuning) and follows a cosine annealing schedule. All sub-networks utilize lightweight U-shaped architectures with self-attention (Zhou et al., 2023) to capture multi-scale features, while the embedded physical models serve as differentiable constraints to stabilize optimization and improve the identifiability of physical parameters.

*Table 1.* Quantitative evaluation on the SRPG dataset. Results are measured in PSNR (dB) and SSIM. **Red** and **blue** indicate the best and second-best performance, respectively.

| Metrics | SLP | PLC | LFP | PD-Net | Dehamer | Deformer | Selfpromer | SGDN | Ours |
|---|---|---|---|---|---|---|---|---|---|
| $PSNR_{S0}$ | 16.927 | 19.084 | 15.976 | 28.786 | 29.110 | 32.179 | 31.773 | 32.201 | 33.696 |
| $SSIM_{S0}$ | 0.7081 | 0.7613 | 0.6926 | 0.9169 | 0.9116 | 0.9336 | 0.9464 | 0.9510 | 0.9480 |
| $PSNR_{DoLP}$ | 13.151 | 21.288 | 18.762 | 25.108 | 24.046 | 25.170 | 25.513 | 25.398 | 28.803 |
| $SSIM_{DoLP}$ | 0.2320 | 0.4988 | 0.3885 | 0.5777 | 0.6074 | 0.6553 | 0.6704 | 0.6634 | 0.7623 |
| $PSNR_{AoLP}$ | 12.664 | 13.677 | 12.608 | 12.543 | 16.063 | 16.868 | 16.627 | 16.138 | 19.431 |
| $SSIM_{AoLP}$ | 0.1525 | 0.2615 | 0.1695 | 0.1914 | 0.3693 | 0.4133 | 0.3986 | 0.3896 | 0.5011 |

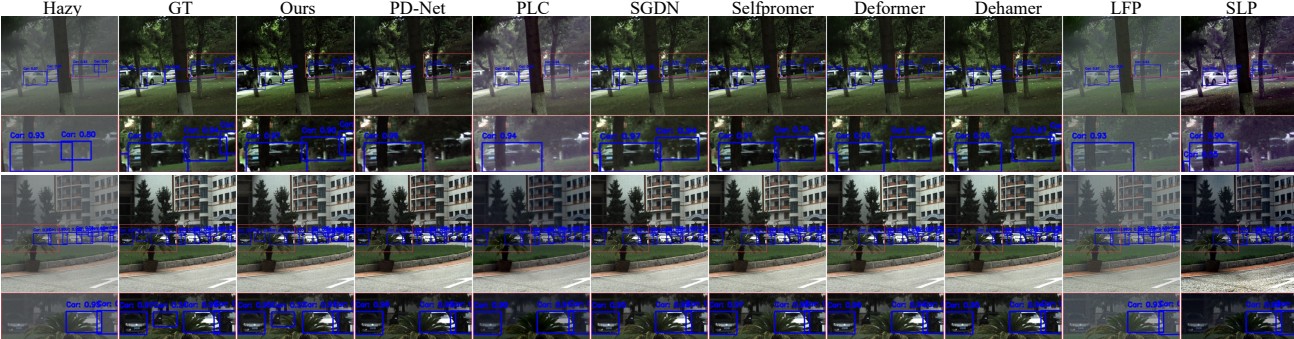

*Figure 5.* Vehicle detection results using various restoration methods. Our approach enhances PCDNet (Lyu et al., 2024) detection via physically consistent polarimetric recovery. *Best viewed by zooming in.*

### 4.3. Evaluation

**Quantitative Evaluation.** We evaluate the restoration performance using standard metrics, including Peak Signal-to-Noise Ratio (PSNR) and Structural Similarity (SSIM). Critically, since the Angle of Linear Polarization (AoLP) is defined on a periodic domain $[0, \pi)$, standard PSNR fails to capture the wrap-around nature of phase errors. To address this, we adopt the periodic PSNR metric (Su et al., 2025), which provides a more mathematically rigorous assessment of AoLP fidelity. As summarized in Table. 1, while DuRP achieves performance comparable to SOTA single-image dehazing models in radiance ($S_0$) restoration, it demonstrates a commanding lead in polarimetric recovery. Specifically, our method yields an improvement of approximately 3 dB in both DoLP and AoLP PSNR. This disparity arises because physics-agnostic models lack the cross-channel constraints required to resolve the complex coupling between scattering and polarization. Since PD-Net (Zhou et al., 2021) relies on the simplifying assumption that airlight and scene AoLP are identical, it exhibits significant estimation bias in general outdoor scenarios; thus, we use the hazy input as its proxy for evaluation. These results indicate that our physics-embedded framework effectively resolves the latent polarimetric state even under scattering.

**Qualitative Evaluation.** Visual comparisons on synthetic

scenes are presented in Figure. 4[5]. Existing single-image dehazing methods, when applied to independent polarization channels, frequently introduce non-physical artifacts and fail to maintain the global structural consistency of the polarization field. In contrast, DuRP produces more accurate and physically plausible reconstructions, effectively preserving the delicate phase relationships in the AoLP maps. By explicitly modeling the orientation-dependent degradation, our method demonstrates a superior capability to recover high-fidelity polarization cues in hazy environments.

### 4.4. Evaluation on Downstream Applications

To further demonstrate the practical benefits of accurately recovering polarization information, we conduct validation on two representative downstream polarization-based vision tasks: polarization-based Vehicle Detection (Dong et al., 2024) and Shape-from-Polarization (Lyu et al., 2024). These tasks rely heavily on the Degree of Linear Polarization (DoLP) and Angle of Linear Polarization (AoLP) cues, which are typically severely degraded by scattering.

**vehicle detection.** We adopt the state-of-the-art polarization-based vehicle detection method, PCDNet (Lyu et al., 2024), for validation, with results shown in Figure. 5. As the results

---

[5]See appendix D.3 and D.4 for extended experiment results.

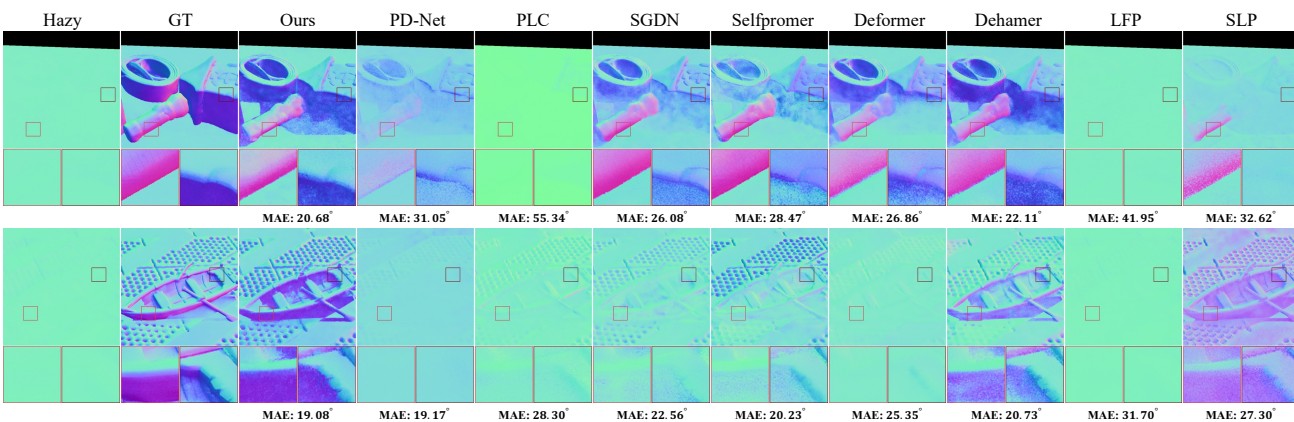

*Figure 6.* Comparison of SfP (Lyu et al., 2024) results across restoration frameworks. Mean Angular Error (MAE ↓) values are indicated for each normal map. Our approach provides superior geometric cues for fine-grained surface reconstruction. *Zoom in for details.*

*Table 2.* Quantitative evaluation results of ablation study on SRPG dataset. Red and blue indicate the best and the second-best performance.

| Settings | PSNR(S0) | SSIM(S0) | PSNR(DoLP) | SSIM(DoLP) | PSNR(AoLP) | SSIM(AoLP) |
|---|---|---|---|---|---|---|
| w/o revised model | 28.244 | 0.8900 | 25.594 | 0.6156 | 12.543 | 0.1914 |
| w/o physical model | 31.670 | 0.9281 | 24.355 | 0.5823 | 16.242 | 0.3551 |
| w/o refined network | 31.652 | 0.9303 | 28.142 | 0.7423 | 18.662 | 0.4840 |
| Ours | 33.696 | 0.9480 | 28.803 | 0.7623 | 19.431 | 0.5011 |

*Table 3.* Inference efficiency under different acceleration settings on the SRPG dataset. For latency, lower is better; for FPS and restoration metrics, higher is better.

| Metrics | FP32 (Base) | FP16 (AMP) | FP16+Comp. |
|---|---|---|---|
| Latency (ms) | 59.6 | 37.8 | 19.0 |
| FPS | 16.8 | 26.4 | 52.6 |
| PSNR($S_0$) | 33.696 | 33.694 | 24.342 |
| SSIM($S_0$) | 0.9480 | 0.9480 | 0.8228 |
| PSNR(DoLP) | 28.803 | 28.803 | 21.129 |
| SSIM(DoLP) | 0.7623 | 0.7622 | 0.5690 |
| PSNR(AoLP) | 19.431 | 19.433 | 16.213 |
| SSIM(AoLP) | 0.5011 | 0.5011 | 0.3624 |

indicate, PCDNet fails to detect all vehicles under hazy conditions. In contrast, when using our restored radiance and polarization information as input, its detection performance is significantly improved. Compared to other methods, our restoration results yield the highest detection accuracy and overall performance.

**Shape from Polarization.** We choose the SOTA shape from polarization (SfP) SfPUEL (Lyu et al., 2024) for validation. Comparisons of the estimated normal maps are shown in Figure. 6[6]. Evidently, our method significantly enhances the accuracy of normal vector estimation in hazy conditions,

clearly surpassing competing methods. By effectively recovering polarization and radiance information, our approach achieves optimal performance, providing the most accurate reconstruction of surface normals.

### 4.5. Ablation Study

To thoroughly evaluate the effectiveness of each module in our proposed method, we conducted ablation experiments on the SRPG dataset.

**Revised Physical Model Validity.** We first assessed the impact of our revised polarization model by replacing it with the traditional model. The results, shown in the "w/o revised model" column of Table 2, demonstrate that our model outperforms the traditional model in recovering both DoLP and radiance information (Since the traditional model assumes no degradation of the scene's AoLP, we use the AoLP of the haze scene instead of the recovered AoLP).

**Embedding Physical Models into Networks.** To verify the importance of parameter estimation based on the physical model, we compared our approach with direct scene polarization prediction. As seen in the "w/o physical model" column of Table 2, the direct prediction strategy significantly underperforms in recovering the polarization. This highlights the difficulty of directly estimating polarization compared to recovering physical model parameters.

**Refinement Network Validity.** We also evaluated the ne-

---

[6]Additional results on Vehicle Detection and SfP are provided in the appendix E.

cessity of the refinement network for accurate recovery of both polarization and radiance information. The "w/o refine network" results in Table 2 show a noticeable performance degradation when the refinement network is omitted. This decline is primarily due to the inherent errors in network-estimated parameters, which, without subsequent refinement, propagate and substantially reduce the accuracy of the recovered scene information.

These ablation results validate the effectiveness of each design choice in our framework. The revised polarization model improves physical fidelity, the physics-guided formulation is essential for accurate polarization recovery, and the refinement network suppresses error propagation. Additional ablation studies are provided in the appendix D.6.

### 4.6. Inference Efficiency

We evaluate the deployment efficiency of **DuRP** under three inference settings (Table 3). Baseline FP32 inference achieves 16.8 FPS. Enabling FP16 automatic mixed precision (AMP) nearly doubles throughput to 26.4 FPS with negligible accuracy loss, making it our recommended deployment setting. Combining FP16 with `torch.compile` further raises speed to 52.6 FPS but incurs consistent metric degradation, so we exclude it from our default configuration. All measurements are conducted on a single RTX 4090. Closing the gap to real-time ($>$30 FPS) on more constrained hardware remains an avenue for future model-level optimization.

## 5. Conclusion

This paper addresses the challenge of recovering accurate polarization in hazy environments. We propose a neural network framework embedded with generalized polarization models, which estimates physical parameters and refines reconstruction results for improved accuracy. In addition, we introduce a novel synthesis pipeline for polarized hazy scenes and construct a corresponding dataset. Experimental results show that our method effectively restores polarization information and significantly improves polarization-based vision tasks, such as vehicle detection and SfP.

**Limitations.** Our method is built upon generalized polarization models derived from the atmospheric scattering model. As a result, its performance may degrade in scenes where the haze does not conform to this model. Additionally, under extreme degradation (e.g., dense fog), where the scene's polarization signal approaches zero, the network may fail to extract meaningful polarization features, leading to inaccurate reconstruction. Representative failure cases and detailed analysis are provided in the appendix material.

## Acknowledgement

We thank all reviewers for their valuable suggestions. This work is supported by the LiaoNing Revitalization Talents Program under Grant XLYC 2502011 and the Nation Key R&D Program of China under Grant 2024YFE0115500.

## Impact Statement

This paper develops a physics-guided learning paradigm to enhance visual reliability in challenging weather. The advancement of such robust sensing capabilities is critical for the safe deployment of autonomous systems. We do not foresee any specific societal or ethical concerns that warrant dedicated discussion beyond those inherent to the field of computer vision.

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

# Appendix

## Appendix Contents

- **A. Comprehensive Mathematical Derivation of the Polarimetric Degradation Model**

- **B. Mathematical Foundations of the Phasor Representation**

- **C. Theoretical Analysis and Proofs**

- **D. Experiment Details**

- **E.Results on Downstream Applications**

---

## A. Comprehensive Mathematical Derivation of the Polarimetric Degradation Model

In this section, we provide the detailed, step-by-step mathematical derivation of the generalized polarimetric degradation equations (Eq. (12) and Eq. (13)) presented in the main paper. Our derivation is grounded in the fundamental principles of Stokes polarimetry and the incoherent superposition of light in scattering media.

### A.1. Polarization Invariance Under Atmospheric Attenuation

We first establish the relationship between the attenuated scene radiance (the direct transmission component $\mathbf{D}$) and the original latent scene radiance $\mathbf{R}$. According to the atmospheric scattering model, the intensity $D$ is related to $R$ via $D = R \cdot t$, where $t \in [0, 1]$ is the transmission coefficient. In standard dehazing formulations, $t$ is typically assumed to be polarization-independent. Consequently, the Stokes parameters of the attenuated light satisfy:

$$S_0^D = S_0^R t, \quad S_1^D = S_1^R t, \quad S_2^D = S_2^R t. \tag{26}$$

The Degree of Linear Polarization (DoLP) $P_D$ and the Angle of Linear Polarization (AoLP) $\theta_D$ of the attenuated light are derived as:

$$
\begin{aligned}
P_D &= \frac{\sqrt{(S_1^D)^2 + (S_2^D)^2}}{S_0^D} = \frac{\sqrt{(S_1^R t)^2 + (S_2^R t)^2}}{S_0^R t} = \frac{t\sqrt{(S_1^R)^2 + (S_2^R)^2}}{t S_0^R} = P_R, \\
\theta_D &= \frac{1}{2}\arctan\left(\frac{S_2^D}{S_1^D}\right) = \frac{1}{2}\arctan\left(\frac{S_2^R t}{S_1^R t}\right) = \frac{1}{2}\arctan\left(\frac{S_2^R}{S_1^R}\right) = \theta_R.
\end{aligned}
\tag{27}
$$

**Remark:** This confirms that atmospheric attenuation acts as a scalar operator on the Stokes vector, leaving the intrinsic polarimetric signatures $(P, \theta)$ invariant. Hence, $P_D$ and $\theta_D$ serve as proxies for the scene's latent polarization state in subsequent derivations.

### A.2. Incoherent Superposition of Stokes Vectors

The observed hazy image $\mathbf{I}$ is the incoherent sum of the attenuated direct transmission $\mathbf{D}$ and the atmospheric airlight $\mathbf{A}$. For incoherent light sources, the total Stokes parameters are additive:

$$S_j^I = S_j^D + S_j^A, \quad j \in \{0, 1, 2\}. \tag{28}$$

Using the relationship $S_1^i = S_0^i P_i \cos(2\theta_i)$ and $S_2^i = S_0^i P_i \sin(2\theta_i)$, we can expand Eq. (28) for the observed linear components as:

$$S_1^I = DP_D \cos(2\theta_D) + AP_A \cos(2\theta_A),$$
$$S_2^I = DP_D \sin(2\theta_D) + AP_A \sin(2\theta_A). \tag{29}$$

where $D = S_0^D$ and $A = S_0^A$ represent the intensities of the scene and airlight, respectively.

### A.3. Derivation of the Observed DoLP (Proof of Eq. 12)

To derive the observed DoLP $P_I$, we compute the magnitude of the combined linear Stokes components:

$$
\begin{aligned}
(IP_I)^2 &= (S_1^I)^2 + (S_2^I)^2 \\
&= (S_1^D + S_1^A)^2 + (S_2^D + S_2^A)^2 \\
&= \underbrace{(S_1^D)^2 + (S_2^D)^2}_{(DP_D)^2} + \underbrace{(S_1^A)^2 + (S_2^A)^2}_{(AP_A)^2} + 2(S_1^D S_1^A + S_2^D S_2^A).
\end{aligned}
\tag{30}
$$

Substituting the expansions from Eq. (29) into the cross-term:

$$
\begin{aligned}
S_1^D S_1^A + S_2^D S_2^A &= (DP_D \cos 2\theta_D)(AP_A \cos 2\theta_A) + (DP_D \sin 2\theta_D)(AP_A \sin 2\theta_A) \\
&= DP_D AP_A \left( \cos 2\theta_D \cos 2\theta_A + \sin 2\theta_D \sin 2\theta_A \right).
\end{aligned}
\tag{31}
$$

Applying the trigonometric identity $\cos(2\theta_D - 2\theta_A) = \cos 2\theta_D \cos 2\theta_A + \sin 2\theta_D \sin 2\theta_A$:

$$S_1^D S_1^A + S_2^D S_2^A = DP_D AP_A \cos 2(\theta_D - \theta_A). \tag{32}$$

By substituting Eq. (32) back into Eq. (30) and employing the algebraic identity $a^2 + b^2 = (a + b)^2 - 2ab$:

$$
\begin{aligned}
(IP_I)^2 &= (DP_D)^2 + (AP_A)^2 + 2DP_D AP_A \cos 2(\theta_D - \theta_A) \\
&= (DP_D + AP_A)^2 - 2DP_D AP_A + 2DP_D AP_A \cos 2(\theta_D - \theta_A) \\
&= (DP_D + AP_A)^2 + 2DP_D AP_A \left[ \cos 2(\theta_D - \theta_A) - 1 \right].
\end{aligned}
\tag{33}
$$

Taking the square root and defining $\xi = \cos 2(\theta_D - \theta_A) - 1$, we arrive at Eq. 12:

$$IP_I = \sqrt{(DP_D + AP_A)^2 + 2DP_D AP_A \xi}. \tag{34}$$

### A.4. Derivation of the Observed AoLP (Proof of Eq. 13)

The observed AoLP $\theta_I$ is defined by the orientation of the combined Stokes vector in the $(S_1, S_2)$ plane:

$$\tan(2\theta_I) = \frac{S_2^I}{S_1^I} = \frac{S_2^D + S_2^A}{S_1^D + S_1^A}. \tag{35}$$

Substituting the components defined in Eq. (29):

$$\tan(2\theta_I) = \frac{DP_D \sin 2\theta_D + AP_A \sin 2\theta_A}{DP_D \cos 2\theta_D + AP_A \cos 2\theta_A}. \tag{36}$$

Solving for $\theta_I$ yields Eq. 13:

$$\theta_I = \frac{1}{2} \arctan \left( \frac{DP_D \sin 2\theta_D + AP_A \sin 2\theta_A}{DP_D \cos 2\theta_D + AP_A \cos 2\theta_A} \right). \tag{37}$$

### A.5. Physical Insight: DoLP Degradation due to AoLP Discrepancy

The parameter $\xi = \cos 2(\theta_D - \theta_A) - 1$ characterizes the interaction between the scene and airlight polarimetric states. Since $\cos \phi \leq 1$, it follows that $\xi \leq 0$. This inequality provides a crucial physical insight: **whenever the orientation of scene polarization ($\theta_D$) differs from that of the airlight ($\theta_A$), the observed DoLP $P_I$ is strictly lower than the weighted average of the constituent DoLPs.** This "depolarization" effect caused by orientation discrepancy is a key reason why independent channel restoration fails and justifies our joint optimization approach.

## B. Mathematical Foundations of the Phasor Representation

In this section, we provide the theoretical justification for the complex-valued phasor representation employed in the **DuRP** framework. We demonstrate that the intensity-weighted summation of phasors in the complex plane is mathematically equivalent to the transcendental polarimetric degradation models derived in Eqs. (12)-(13).

### B.1. Definition and Continuity

We represent the polarimetric state of a light component $i$ as a complex phasor $\mathcal{C}_i$:

$$\mathcal{C}_i = P_i e^{j2\theta_i} = P_i \cos(2\theta_i) + jP_i \sin(2\theta_i), \tag{38}$$

where $P_i \in [0, 1]$ is the Degree of Linear Polarization (DoLP) and $\theta_i \in [0, \pi)$ is the Angle of Linear Polarization (AoLP). By mapping the periodic AoLP onto a $2\theta$ phase, we ensure that the phasor $\mathcal{C}_i$ remains continuous across the $0/\pi$ boundary, effectively eliminating the phase-jump singularities that typically hinder direct regression of $\theta$.

### B.2. Equivalence Proof via Complex Algebra

Consider the incoherent superposition of the scene component **D** and the airlight **A**. The total observed intensity $I$ and the corresponding phasor $\mathcal{C}_I$ are governed by the intensity-weighted sum:

$$I \cdot \mathcal{C}_I = D \cdot \mathcal{C}_D + A \cdot \mathcal{C}_A. \tag{39}$$

#### B.2.1. PROOF FOR OBSERVED DoLP ($P_I$)

To derive the magnitude $P_I$, we compute the squared modulus of both sides of Eq. (39). Utilizing the identity $|z|^2 = z \cdot z^*$, where $z^*$ is the complex conjugate:

$$
\begin{aligned}
(IP_I)^2 &= |DP_D e^{j2\theta_D} + AP_A e^{j2\theta_A}|^2 \\
&= (DP_D e^{j2\theta_D} + AP_A e^{j2\theta_A})(DP_D e^{-j2\theta_D} + AP_A e^{-j2\theta_A}) \\
&= (DP_D)^2 + (AP_A)^2 + DP_D AP_A \left( e^{j2(\theta_D - \theta_A)} + e^{-j2(\theta_D - \theta_A)} \right).
\end{aligned}
\tag{40}
$$

Applying Euler's identity, $e^{j\phi} + e^{-j\phi} = 2\cos\phi$, Eq. (40) simplifies to:

$$(IP_I)^2 = (DP_D)^2 + (AP_A)^2 + 2DP_D AP_A \cos 2(\theta_D - \theta_A). \tag{41}$$

By adding and subtracting $2DP_D AP_A$ and grouping terms, we obtain:

$$
\begin{aligned}
(IP_I)^2 &= (DP_D + AP_A)^2 - 2DP_D AP_A + 2DP_D AP_A \cos 2(\theta_D - \theta_A) \\
&= (DP_D + AP_A)^2 + 2DP_D AP_A \left[ \cos 2(\theta_D - \theta_A) - 1 \right],
\end{aligned}
\tag{42}
$$

which is identically the expression for observed DoLP in Eq. (13).

#### B.2.2. PROOF FOR OBSERVED AoLP ($\theta_I$)

The resulting AoLP $\theta_I$ is derived from the argument of the complex sum in Eq. (39):

$$2\theta_I = \arg(I\mathcal{C}_I) = \arg \left( \sum_{i \in \{D,A\}} S_0^i P_i \cos 2\theta_i + j \sum_{i \in \{D,A\}} S_0^i P_i \sin 2\theta_i \right). \tag{43}$$

The argument of a complex number $x + jy$ is given by $\arctan(y/x)$. Thus:

$$2\theta_I = \arctan \left( \frac{DP_D \sin 2\theta_D + AP_A \sin 2\theta_A}{DP_D \cos 2\theta_D + AP_A \cos 2\theta_A} \right). \tag{44}$$

Dividing by 2 yields the exact form of Eq. (14), concluding the proof.

## B.3. Implications for Deep Learning

The mapping of polarimetric physics into the complex domain provides a powerful **inductive bias** for our network. While the relationships in Eqs. (12)-(13) are highly non-linear and coupled in the real domain, they become simple linear operations (vector additions) in the phasor space. This linearization facilitates a more stable gradient flow during backpropagation and ensures that the **DuRP** framework implicitly respects the laws of incoherent light superposition, even when learning from complex, high-dimensional sensory data.

# C. Theoretical Analysis and Proofs

In this section, we provide a detailed theoretical foundation for our generalized polarimetric model and the coordinate-independent features used in our framework.

## C.1. Theoretical Error Analysis: Generalized vs. Traditional Models

This subsection quantifies the systematic bias inherent in traditional polarimetric dehazing models and justifies the necessity of our generalized approach.

**The Traditional Model as a Singular Special Case.** Most prior polarimetric dehazing methods (e.g., (Zhou et al., 2021; Yang et al., 2025)) rely on a simplified assumption that the airlight polarization orientation $\theta_A$ is identical to that of the scene radiance $\theta_D$. Under this condition, the orientation discrepancy $\Delta\theta = |\theta_D - \theta_A|$ vanishes, implying $\xi = \cos 2(0) - 1 = 0$. Consequently, the observed DoLP simplifies to a basic intensity-weighted average:

$$P_I^{\text{trad}} = \frac{DP_D + AP_A}{D + A}. \tag{45}$$

This scalar addition treats polarization as a non-vectorial quantity. However, in diverse outdoor environments, the complex interplay between solar geometry, object surface normals, and scattering media ensures that $\theta_A \neq \theta_D$ is the near-universal rule rather than the exception.

**Quantification of Depolarization Error.** When $\theta_A \neq \theta_D$, the physical interaction between the scene and airlight introduces a negative term $2DP_DAP_A\xi \leq 0$, where $\xi = \cos 2(\Delta\theta) - 1$. The true observed DoLP, $P_I^{\text{true}}$, is governed by:

$$P_I^{\text{true}} = \frac{1}{I}\sqrt{(DP_D + AP_A)^2 + 2DP_DAP_A\xi}. \tag{46}$$

Since $\xi \leq 0$, it follows that $P_I^{\text{true}} < P_I^{\text{trad}}$. The systematic bias $\mathcal{E} = P_I^{\text{trad}} - P_I^{\text{true}}$ can be approximated via first-order Taylor expansion $\sqrt{1-x} \approx 1 - x/2$:

$$\mathcal{E} \approx \frac{2DP_DAP_A\sin^2(\Delta\theta)}{I(DP_D + AP_A)}. \tag{47}$$

This depolarization error $\mathcal{E}$ is proportional to $\sin^2(\Delta\theta)$ and the product of intensities $D \cdot A$. The traditional model consistently overestimates the observed polarization magnitude. In regions with dense scattering or complex geometry, this leads to "polarimetric over-compensation," failing to account for the destructive interference between orientation-mismatched phasors.

**Impact on Restoration.** In the **DuRP** framework, such overestimation in $P_I$ would propagate through the Radiance Reconstruction Model (RRM), leading to non-physical radiance values and chroma distortion. By explicitly modeling $\Delta\theta \neq 0$, our generalized approach eliminates this intrinsic model misspecification error, ensuring physically consistent restoration even under complex scattering geometries.

## C.2. Mathematical Proof of Rotation Invariance

This subsection proves that the max and min intensity representations $(I_{\max}, I_{\min})$ are invariant under the rotation of the observer's reference frame.

**Transformation of Stokes Components.** Consider a rotation of the camera or polarizer reference frame by an angle $\phi$. The linear Stokes components $[S_0, S_1, S_2]^{\mathrm{T}}$ transform according to the following Mueller matrix relationship:

$$\begin{aligned}
S_0' &= S_0, \\
S_1' &= S_1 \cos(2\phi) + S_2 \sin(2\phi), \\
S_2' &= -S_1 \sin(2\phi) + S_2 \cos(2\phi).
\end{aligned} \tag{48}$$

**Proof for Intensity Extrema.** To establish the invariance of $I_{\max}$ and $I_{\min}$, we examine the quadratic term $L = S_1^2 + S_2^2$ within the definition. After rotation, the transformed term $L'$ is given by:

$$\begin{aligned}
L' &= (S_1')^2 + (S_2')^2 \\
&= (S_1 \cos 2\phi + S_2 \sin 2\phi)^2 + (-S_1 \sin 2\phi + S_2 \cos 2\phi)^2 \\
&= S_1^2(\cos^2 2\phi + \sin^2 2\phi) + S_2^2(\sin^2 2\phi + \cos^2 2\phi) \\
&\quad + 2S_1 S_2 \sin 2\phi \cos 2\phi - 2S_1 S_2 \sin 2\phi \cos 2\phi \\
&= S_1^2 + S_2^2 = L.
\end{aligned} \tag{49}$$

Since $S_0$ is intrinsically invariant and we have proved $L' = L$, it follows that the linear combinations $I_{\max/\min} = \frac{1}{2}(S_0 \pm \sqrt{L})$ are invariant under any rotation $\phi$. This ensures that $(I_{\max}, I_{\min})$ provide coordinate-independent physical descriptors, which serve as a robust inductive bias for the network. $\square$

## D. Experiment Details

### D.1. Details of dataset generation.

To bridge the gap in paired polarimetric hazy data, we develop a physically-grounded synthesis pipeline to construct the SRPG dataset. For each clear scene, we stochastically sample the atmospheric light $A_\infty \sim \mathcal{U}(0.85, 0.95)$ and the scattering coefficient $\beta \sim \mathcal{U}(0.0025, 0.0035)$, enforcing a spectral ordering (i.e., $A_{\infty,r} < A_{\infty,g} < A_{\infty,b}$ and $\beta_r < \beta_g < \beta_b$) to mimic real-world Rayleigh scattering dynamics. Utilizing the normalized scene depth $z$, the transmission map is computed as $t(x) = \mathrm{clip}(\exp(-\beta \cdot z(x)), 0.1, 1.0)$. We simulate the airlight components $A_\alpha$ at polarizer angles $\alpha \in \{0°, 45°, 90°, 135°\}$ according to the generalized polarimetric model: $A_\alpha = \frac{1}{2} A_\infty(1-t)[1 + P_A \cos(2(\alpha - \theta_A))]$, where the airlight degree of polarization $P_A$ and angle of polarization $\theta_A$ are sampled from $\mathcal{U}(0.05, 0.4)$ and $\mathcal{N}(\mu, \sigma)$, respectively. The attenuated scene component is obtained via $D_\alpha = R_\alpha \cdot t$. Finally, the hazy intensities are synthesized as $I_\alpha = \mathrm{clip}(\mathcal{N}(D_\alpha + A_\alpha, \sigma_n), 0, 1)$ with $0.5\%$ additive Gaussian noise to account for sensor non-idealities. This pipeline, with an enlargement factor of 3, yields 1,161 training pairs from 387 outdoor scenes, providing a rich diversity of haze densities and polarimetric signatures for robust network supervision.

*Table 4.* Summary of Training Hyperparameters

| Hyperparameter | Value |
|---|---|
| Optimizer | Adam ($\beta_1 = 0.9, \beta_2 = 0.999, \epsilon = 10^{-8}$) |
| Initial Learning Rate (P/R-Stage) | $1 \times 10^{-4}$ |
| Initial Learning Rate (Full-Stage) | $1 \times 10^{-5}$ |
| Learning Rate Scheduler | Cosine Annealing ($\eta_{\min} \in [10^{-5}, 10^{-6}]$) |
| T_max (for Cosine Schedule) | 32 |
| Weight Decay | $1 \times 10^{-4}$ |
| Patch Size | $512 \times 512$ |
| Training Epochs | 400 |

### D.2. Implementation Details

In this section, we provide exhaustive details regarding the network implementation, dataset preparation, and the multi-stage training protocol to ensure the reproducibility of the results presented in the main paper.

**Hardware and Software Environment**    All models were implemented using the **PyTorch** framework and trained on a Linux-based workstation equipped with an **NVIDIA RTX 4090 GPU** (24GB VRAM).

**Training Protocol and Hyperparameters**    We employed a robust optimization configuration to stabilize the estimation of physical parameters within the **DuRP** framework. The primary hyperparameters are summarized in Table 4.

**Multi-Stage Optimization Strategy**    The training of the **DuRP** framework is partitioned into three distinct stages to decouple the physical variables and ensure stable convergence:

1. **P-Stage (Polarization Reconstruction):** The **PRNet** is trained to recover the polarimetric phasor components $(K, C_A, C_D)$. We use a batch size of 1 and a learning rate of $10^{-4}$, focusing on minimizing the polarimetric loss $\mathcal{L}_P$ to establish a physically consistent foundation.

2. **R-Stage (Radiance Reconstruction):** Independently, the **IRNet** is trained to recover the scene radiance $R$ and atmospheric light $A_\infty$. This stage emphasizes structural restoration and color consistency, utilizing a larger batch size of 4.

3. **Joint Fine-tuning (Full Stage):** The weights from the pre-trained PRNet and IRNet are loaded as the foundation. The entire framework is then optimized end-to-end. During this stage, the learning rate is reduced to $10^{-5}$ and the minimum eta ($\eta_{\min}$) is set to $10^{-6}$ for fine-grained convergence. We utilize a `CosineAnnealingLR` policy with $T_{\max} = 32$ to facilitate escape from local minima.

**Evaluation Protocol**    During the evaluation phase, images were processed at their full resolution. We utilized both standard image restoration metrics (PSNR, SSIM) and the specialized **periodic PSNR** for AoLP assessment to identify the optimal checkpoints for each stage.

### D.3. Simulation experiment results

In this section, we provide a comprehensive visual evaluation to further substantiate the generalizability and robust performance of the **DuRP** framework. We present an expanded set of qualitative comparisons against a diverse suite of baselines, including: **Polarization-based methods:** PD-Net (Zhou et al., 2021) and PLC (Su et al., 2024), which represent the current paradigm in leveraging polarimetric priors for dehazing. **Single-image intensity-based methods:** Six SOTA frameworks, namely SGDN (Fang et al., 2025), Selfpromer (Wang et al., 2024a), Deformer (Song et al., 2023), Dehamer (Guo et al., 2022), LFP (Li & Zhang, 2024), and SLP (Ling et al., 2023). The exhaustive results, visualized in Fig. 11, serve as a critical extension to the findings discussed in the main paper.

To provide an intuitive yet physically rigorous representation of the polarimetric recovery, we render the Degree of Linear Polarization (DoLP, $p$) and the Angle of Linear Polarization (AoLP, $\theta$) as normalized colormaps. Given that the input data contains multi-spectral polarimetric information, these images are computed by averaging the recovered Stokes parameters across the RGB channels. As evidenced by the results, our method demonstrates significant advantages. (i) While single-image dehazing methods (e.g., *Dehamer*, *SLP*) are designed to restore radiance ($S_0$), they exhibit a total failure in maintaining the polarization state coupling. Even for the polarization-based baselines, *PD-Net* frequently introduces non-physical artifacts in the AoLP maps due to its simplistic assumption of orientation invariance. In contrast, **DuRP** effectively resolves the latent phasors, producing smooth and physically consistent AoLP maps even in dense haze regions. (ii) Many baselines suffer from chromatic distortion when processing hazy inputs with spatially variant scattering coefficients. By embedding the generalized physical model into the network architecture, our framework ensures that the restored radiance and polarization are coupled through deterministic optical laws, effectively eliminating the color casts prevalent in *LFP* and *PLC*. (iii) In scenarios with extreme haze density (low transmission $t$), baseline methods often lose textural detail, resulting in over-smoothed or noisy reconstructions. However, **DuRP** maintains more details in the DoLP map, which is crucial for downstream geometric tasks like SfP.

### D.4. Real experiment results

To further validate the robustness of the proposed **DuRP** framework, we conduct qualitative comparisons on diverse real-world scenes, ranging from indoor scene to outdoor cityscapes, as illustrated in Figure. 7

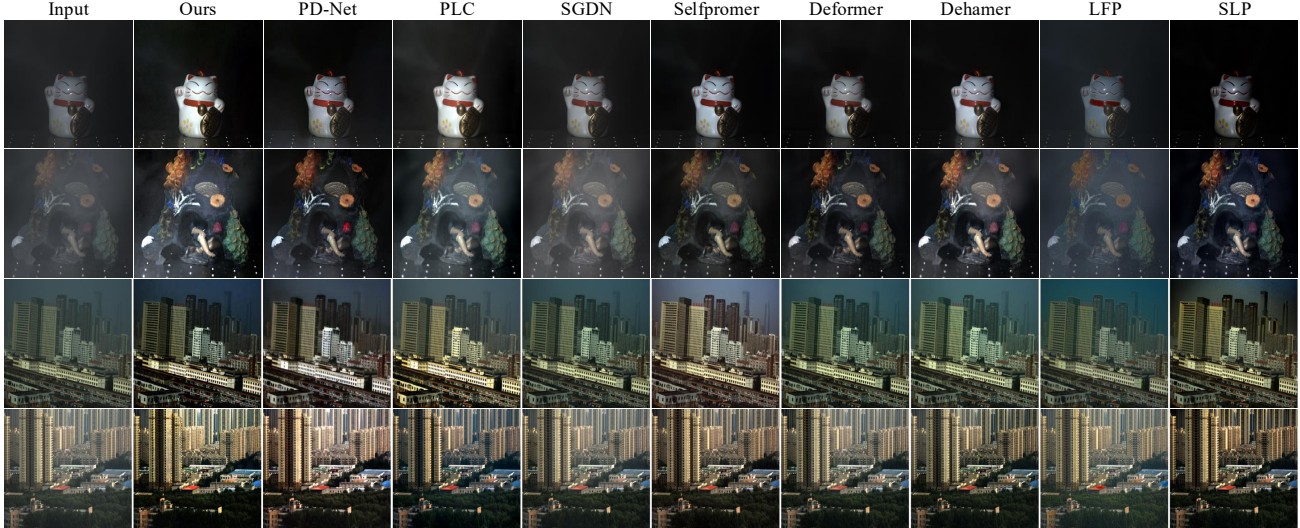

*Figure 7.* **Qualitative comparison on real-world hazy scenes.** We evaluate the performance of DuRP against state-of-the-art single-image dehazing and polarimetric restoration methods across diverse indoor and outdoor environments. Dehazing results are brightness-adjusted for better visualization.

**Evaluation Constraints and Metric Selection.** A fundamental challenge in real-world polarimetric dehazing is the inherent unavailability of ground-truth (GT) polarimetric information. In uncontrolled environments, the intrinsic Degree of Linear Polarization (DoLP) and Angle of Linear Polarization (AoLP) of the clear scene cannot be measured simultaneously with the hazy observation. Consequently, quantitative polarimetric metrics (e.g., periodic PSNR) are inapplicable. To ensure a rigorous assessment, we primarily visualize the **restored radiance** ($S_0$) as a observable proxy. The visual quality of $S_0$ serves as a direct reflection of the accuracy of the underlying physical parameter estimation; any significant bias in the predicted polarimetric phasors would inevitably manifest as color casts or structural artifacts in the final radiance output.

**Comparative Analysis of Restoration Quality.** As shown in Figure. 7, our method produces visually consistent restoration results on real-world hazy scenes. In the indoor cases (Rows 1–2), several competing methods either preserve residual haze or introduce noticeable color deviations, resulting in grayish, or over-enhanced appearances. In comparison, **DuRP** improves the visibility of the main objects while maintaining relatively natural color and contrast. In the outdoor scenes (Rows 3–4), haze removal becomes more challenging due to the depth-varying scattering in distant buildings and urban backgrounds. Existing methods often leave visible haze, or cause color bias. Our method achieves a better balance between haze suppression and appearance preservation, recovering clearer structures without introducing obvious artifacts. These results indicate that the proposed physics-embedded design generalizes to real-world hazy conditions.

### D.5. Failure-Case analysis under dense fog

Under extremely dense fog, the direct scene component is severely attenuated, making the observed polarization cues weak and spatially incomplete. We show several failure cases in real dense-fog scenes in Fig. 8. These examples illustrate that the degradation is closely related to the loss of measurable polarization cues in the input. In the first row, the DoLP map contains almost no reliable information on the left side, while only the lower-right region still preserves visible polarization responses. As a result, our method removes haze more effectively in the lower-right region but leaves substantial residual haze on the left side. In the second row, the DoLP information in the upper-left region is almost completely lost, which leads to visible distortion in the corresponding restored region. In contrast, regions with more reliable polarization responses are restored more successfully, and most of the haze is removed.

These observations suggest that dense fog poses a fundamental challenge for polarization-based restoration methods. DuRP is still constrained by the amount of physically measurable scene information present in the input. When the scene signal is nearly absent, neither the learned prior nor the embedded physical model can fully recover accurate radiance and polarization information. We include this analysis to clarify the limitation of DuRP under extreme degradation and to avoid overstating its generalization capability in all real-world scattering conditions.

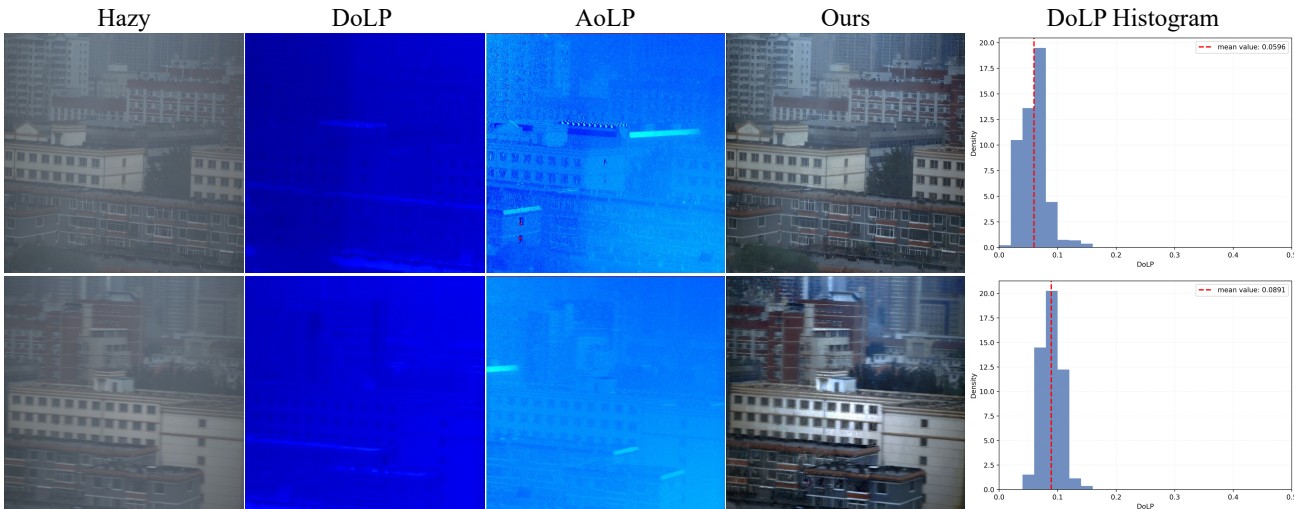

*Figure 8.* **Failure-case analysis under dense fog.** We show representative real-world examples where restoration performance degrades under extremely dense haze.

*Table 5.* Ablation on physics-guided polarization formulation. Red and blue indicate the best and second-best performance, respectively.

| Settings | PSNR($S_0$) | SSIM($S_0$) | PSNR(DoLP) | SSIM(DoLP) | PSNR(AoLP) | SSIM(AoLP) |
|---|---|---|---|---|---|---|
| w/o polar. physics | 32.010 | 0.9330 | 25.640 | 0.6100 | 16.780 | 0.3110 |
| w/o all physics | 31.670 | 0.9280 | 24.360 | 0.5820 | 16.240 | 0.3550 |
| Ours | 33.700 | 0.9480 | 28.800 | 0.7620 | 19.430 | 0.5010 |

## D.6. Ablation experiment results

In this section, we provide additional ablation studies, focusing on the necessity of the physics-guided formulation and the design choice of the backbone in AL-Net.

**Effect of Physics-Guided Polarization Formulation.** We first conduct a detailed ablation to evaluate the impact of removing the physical modeling in polarization recovery. Specifically, we consider two variants: (i) removing only the polarization physical model in the P-Stage ("w/o polar. physics"), and (ii) removing all physical models and directly predicting the target signals ("w/o all physics"). The quantitative results are summarized in Table 5. Compared with our full model, removing the polarization physical model leads to a significant performance drop, with DoLP PSNR decreasing by 3.16 dB and AoLP PSNR decreasing by 2.65 dB. When all physical models are removed, the degradation becomes even more pronounced, with DoLP and AoLP PSNR dropping by 4.44 dB and 3.19 dB, respectively. These results indicate that direct prediction struggles to preserve the intrinsic coupling between polarization components, demonstrating the necessity of the physics-guided formulation.

**Backbone Design for Atmospheric Light Estimation.** We further investigate the impact of the backbone architecture in AL-Net. Although atmospheric light estimation benefits from global context, we find that the lightweight U-Net with embedded self-attention achieves a better trade-off between accuracy and efficiency under the current data regime.

To validate this, we replace the U-Net backbone in AL-Net with a Swin-T Transformer. The results are shown in Table 6. The Swin-based variant more than doubles the number of parameters (from 12.2M to 29.9M) and reduces the inference speed from 16.8 FPS to 7.0 FPS. Meanwhile, it also leads to consistent performance degradation across $S_0$, DoLP, and AoLP. We attribute this behavior to the limited scale of high-quality paired polarimetric datasets. In such scenarios, CNN-based models provide stronger inductive biases (e.g., spatial locality), which help prevent overfitting and improve generalization.

*Table 6.* Ablation on the backbone of AL-Net. Red and blue indicate the best and second-best performance, respectively.

| Settings | Params (M) | FPS | PSNR($S_0$) | SSIM($S_0$) | PSNR(DoLP) | SSIM(DoLP) | PSNR(AoLP) | SSIM(AoLP) |
|---|---|---|---|---|---|---|---|---|
| Ours+Swin | 29.9 | 7.0 | 31.701 | 0.9310 | 27.560 | 0.7090 | 18.490 | 0.4430 |
| Ours | 12.2 | 16.8 | 33.700 | 0.9480 | 28.800 | 0.7620 | 19.430 | 0.5010 |

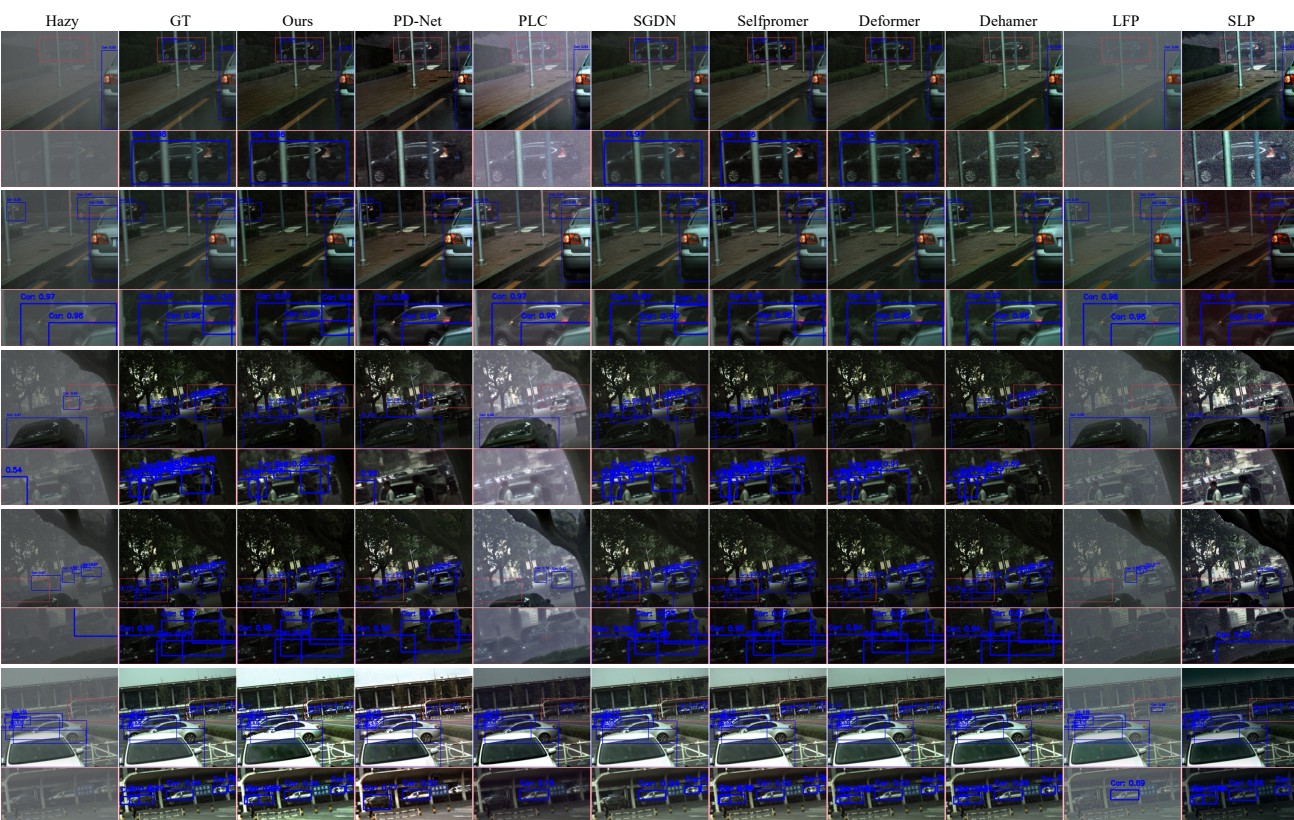

*Figure 9.* Vehicle detection results using restored images from different methods. Our method yields clearer object boundaries and improved detection confidence. *Please zoom in for detailed visual information.*

## E. Results on Downstream Applications

In this section, we further demonstrate that the **DuRP** framework is not merely a visual enhancement tool but a robust pre-processing engine for high-level perception and geometric reconstruction. We evaluate the utility of restored radiance and polarimetric information on two critical downstream tasks: **vehicle detection** and **Shape-from-Polarization (SfP)**.

**Results on Vehicle Detection** Vehicle detection in hazy environments is particularly challenging due to the severe attenuation of the target's polarimetric saliency. For metallic surfaces, the degree of linear polarization (DoLP) provides a vital signature that distinguishes the vehicle from the diffusive scattering background. As illustrated in Fig. 9, we utilize the state-of-the-art *PCDNet* detector to evaluate the quality of the restored inputs. Our DuRP framework achieves the most robust vehicle detection performance among all evaluated methods. Specifically, as shown in the bounding box visualizations and confidence scores in Fig. 9, our approach significantly reduces the miss rate for targets in dense haze.

**Results on Shape from Polarization** Shape-from-Polarization (SfP) is notoriously sensitive to the accuracy of the Angle of Linear Polarization (AoLP, $\theta$), which determines the azimuthal component of surface normals. Any systematic bias in the dehazing process propagates non-linearly to the final 3D reconstruction. Our DuRP framework achieves the most accurate 3D reconstruction results among all evaluated methods. As illustrated by the estimated surface normals and geometric

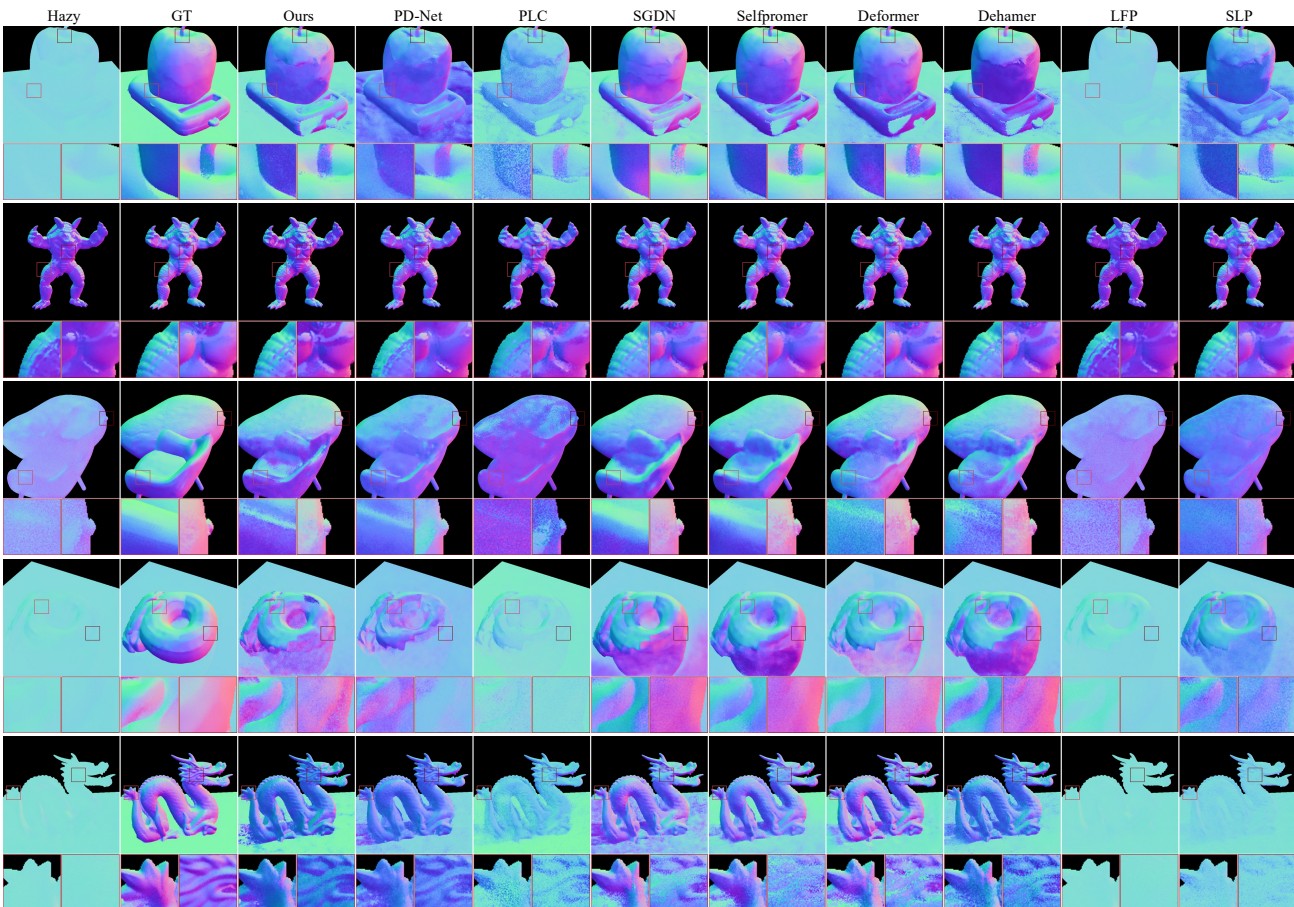

*Figure 10.* Shape-from-polarization (SfP) reconstructions using restored inputs from different methods. Our approach enhances polarization cues, facilitating more accurate 3D surface recovery.

details in Fig. 10, our approach effectively preserves the structural integrity of the scene even under extreme scattering.

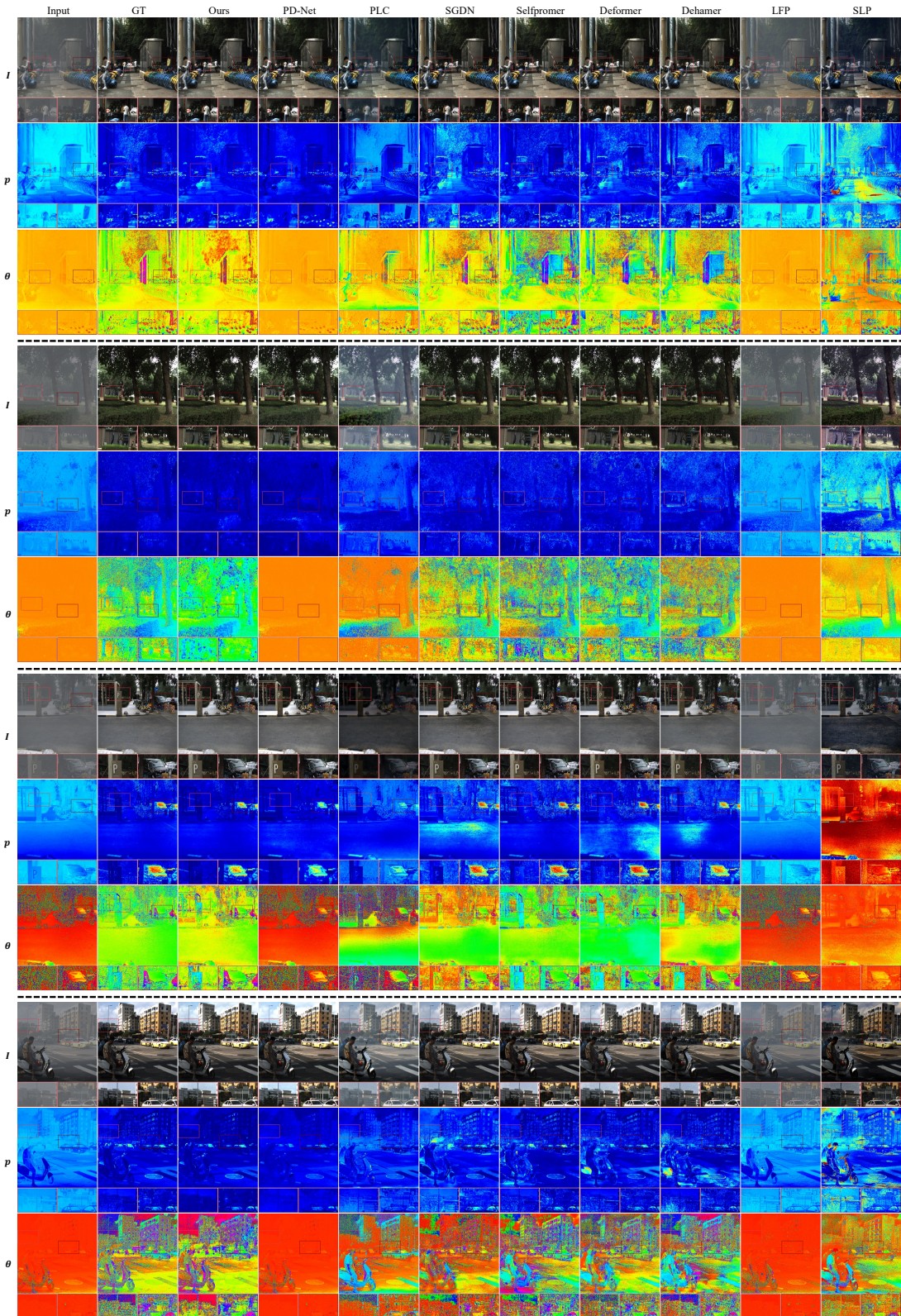

*Figure 11.* Comprehensive visual comparison on synthetic scenes. This figure illustrates the restoration performance of radiance ($S_0$) and polarimetric cues. For visualization, the multi-channel DoLP ($p$) and AoLP ($\theta$) maps are normalized and averaged across the RGB spectrum to generate a unified polarimetric representation. Compared to baselines, our framework effectively eliminates polarization artifacts and maintains physical consistency. *Please zoom in for a detailed inspection of boundaries and phase relationships.*

