# OpenReview forum: "DuRP: Dual-Stage Physics-Embedded Learning for Joint Radiance and Polarization Restoration"
_ICML.cc/2026/Conference — ICML 2026 regular_

### Official Review · Reviewer_L28b · 2026-03-08

**Soundness:** 3
**Presentation:** 3
**Significance:** 3
**Originality:** 3
**Overall Recommendation:** 4
**Confidence:** 3

**Summary:**

The paper restores scene radiance and polarimetric states in haze by correcting the assumption that atmospheric light and scene radiance share identical polarization angles. Contributions include a generalized scattering model using complex phasor representation to linearize equations, the SRPG dataset generated via stochastic sampling, and mathematical derivations proving parameter decouplability from Stokes vectors.

**Compliance With Llm Reviewing Policy:**

Affirmed.

**Final Justification:**

The physical model proposed in the paper has some novelty. However, the dataset constructed based on this model is purely synthetic and lacks sufficient real data support, which limits the credibility of the work. Given the overall novelty of the work, I have decided to maintain my initial score.

**Key Questions For Authors:**

However, its primary weakness lies in the questionable real-world validity of its synthetic SRPG dataset: while the authors employ depth maps and stochastic sampling to simulate complex distributions, the supplementary materials only demonstrate qualitative dehazing results on real images, failing to provide quantitative validation of the recovered polarimetric states  or concrete multi-task applications, thus leaving unproven whether the simulation data effectively translates to real-world scenarios requiring joint restoration.

**Limitations:**

yes

**Strengths And Weaknesses:**

The paper proposes the DuRP framework, which derives a generalized polarimetric scattering model to relax the traditional assumption that the Angle of Linear Polarization of atmospheric light equals that of scene radiance, and introduces a complex phasor representation to transform non-linear couplings into linear operations; its theoretical derivation, grounded in the principles of incoherent superposition of Stokes vectors and Malus's Law, is mathematically rigorous with complete proofs provided in the appendix, lending it high credibility.

However, its primary weakness lies in the questionable real-world validity of its synthetic SRPG dataset: while the authors employ depth maps and stochastic sampling to simulate complex distributions, the supplementary materials only demonstrate qualitative dehazing results on real images, failing to provide quantitative validation of the recovered polarimetric states  or concrete multi-task applications, thus leaving unproven whether the simulation data effectively translates to real-world scenarios requiring joint restoration.

---

> ### Author Rebuttal · Authors · 2026-03-30
>
> We thank you for the insightful comments on the SRPG dataset's real-world validity. This constructive feedback has guided us to significantly strengthen the paper with additional quantitative analyses on sim-to-real transferability.
> > __W1 \& Q1:__ Lack of quantitative validation for real-world scenarios and the sim-to-real gap.
>
> __A1:__ We agree that qualitative visualizations alone are insufficient. However, acquiring pixel-wise polarimetric ground truth in dynamic real hazy scenes is practically impossible. This inherent difficulty is why existing works heavily rely on physics-based simulations alongside limited real-scene validation. To directly address your concern, we provide two rigorous quantitative evaluations on real-world data to prove that our SRPG-trained model effectively transfers to physical scenarios:
>
> Since no established no-reference metrics exist specifically for polarization states, we evaluate the restored radiance ($S_0$). Crucially, in our physics-embedded framework, accurate $S_0$ recovery strictly relies on the correct estimation of polarimetric parameters. Thus, $S_0$ quality serves as a reliable proxy for polarization restoration. Using four standard metrics on real images, DuRP achieves the best overall average rank, indicating consistently superior perceptual quality.
> |Method|NIQE↓|BRISQUE↓|MANIQA↑|DBCNN↑|Avg. Rank↓|
> -|-|-|-|-|-
> |SLP|5.62|24.27|0.325|0.461|7.5|
> |PLC|5.48|24.06|0.327|0.479|6.0|
> |LFP|5.73|21.45|0.345|0.505|4.5|
> |PD-Net|4.23|17.71|0.341|0.527|2.3|
> |Dehamer|5.15|25.62|0.340|0.491|5.3|
> |Deformer|5.71|22.87|0.330|0.475|6.5|
> |Selfpromer|5.39|26.29|0.365|0.534|4.0|
> |SGDN|5.72|25.27|0.338|0.479|6.8|
> |Ours|4.50|21.68|0.348|0.547|2.0|
>
> To further test whether the restored outputs are useful in practice, we manually annotated a real-world vehicle detection data and using the SOTA polarimetric detector, PCDNet[1]. As shown below, oour method achieves the highest Precision and AP. This quantitatively proves that our simulation-trained model effectively translates to real-world scenarios requiring joint restoration—actively benefiting downstream perception rather than merely improving visual appearance.
> | Method | P ↑ | AP↑ | AP50↑ | Avg. Rank ↓ |
> |---|---:|---:|---:|---:|
> | Hazy | 0.255 | 0.110 | 0.017 | 7.0 |
> | SLP | 0.222 | 0.113 | 0.015 | 7.0 |
> | PLC | 0.466 | 0.112 | 0.014 | 6.0 |
> | LFP | 0.333 | 0.132 | 0.031 | 3.3 |
> | Deformer | 0.408 | 0.125 | 0.020 | 3.3 |
> | Selfpromer | 0.499 | 0.140 | 0.028 | 2.0 |
> | SGDN | 0.372 | 0.116 | 0.020 | 4.0 |
> | Ours | 0.500 | 0.146 | 0.020 | 1.3 |
>
> In summary, these auxiliary evaluations provide quantitative evidence for the effectiveness and practical utility of our approach, demonstrating that the model trained on our simulated data possesses the capability to translate to real-world scenarios requiring joint restoration.
>
> [1] Exploiting Polarized Material Cues for Robust Car Detection, AAAI 2024.

---

> > ### Author Rebuttal · Reviewer_L28b · 2026-04-01
> >
> > The authors have provided quantitative comparisons in real-world scenarios, demonstrating that the proposed method is competitive in both dehazing and downstream tasks.  However, since the authors propose a purely simulation-based model built upon a completely new task, without sufficient real-world data to support it, I believe the innovation of this work is limited. I therefore decide to maintain my score.

---

> > > ### Author Response · Authors · 2026-04-02
> > >
> > > Thank you for your time and constructive feedback. We are glad to hear that our previous responses addressed most of your concerns.

---

### Official Review · Reviewer_XBwd · 2026-03-11

**Soundness:** 4
**Presentation:** 3
**Significance:** 4
**Originality:** 4
**Overall Recommendation:** 4
**Confidence:** 4

**Summary:**

A two-stage physical embedding learning framework named DuRP is proposed to address the joint restoration of scene luminance and polarization information in scattering environments such as haze. Addressing the limitations of existing methods—which often neglect polarization consistency or rely on idealized physical assumptions during defogging—the authors derive a more universal generalized polarization physical model and integrate it as an inductive bias into deep networks. By sequentially restoring polarization state and scene luminance, this framework not only significantly enhances fog removal quality but also accurately reconstructs physically meaningful polarization degree and polarization angle. This effectively supports downstream visual applications such as 3D surface reconstruction and object detection.

**Compliance With Llm Reviewing Policy:**

Affirmed.

**Key Questions For Authors:**

1.The sub-networks employ lightweight U-Net architectures, while atmospheric light estimation typically benefits from the long-range global context provided by Transformers. Please provide a brief discussion or ablation study to justify the choice of U-Net over a Transformer backbone, specifically regarding the trade-off between computational efficiency and global dependency capture.

2.Equation 22 lacks a specific mathematical expansion for the complex phasor loss, hindering reproducibility. The authors must clarify if the L1/L2 norms are computed by separating real/imaginary parts or via the complex modulus, as these implementations yield fundamentally different gradients during backpropagation.

3.The manuscript should clarify how the network distinguishes polarization caused by atmospheric scattering from polarization inherent to specific materials (e.g., glass, metal, or water). It is recommended to supplement test cases for such highly reflective materials to verify whether global parameter estimation possesses sufficient physical robustness when confronted with strong local polarization features.

4.The SRPG dataset uses ideal Gaussian noise and lacks absolute physical scale, ignoring complex DoFP sensor artifacts like spatial cross-talk. To address this gap, please supplement the evaluation with a robustness experiment using complex sensor noise patterns to demonstrate stability on real-world physical data.

**Limitations:**

DuRP adopts a dual-stage architecture comprising multiple refinement modules. While this decoupled design enhances accuracy, it also significantly increases computational parameters and inference latency. For hardware platforms demanding high-speed processing—such as UAV obstacle avoidance or autonomous driving—current architectures may require further lightweight pruning or knowledge distillation to meet stringent real-time requirements.

**Strengths And Weaknesses:**

This paper demonstrates solid overall performance. Its derived generalized polarization physical model corrects overly idealized assumptions in traditional methods and effectively couples physical principles with deep learning through a two-stage neural network. Experiments not only cover conventional quality metrics but also rigorously validate the accuracy of physical information recovery through downstream tasks such as 3D reconstruction. The paper maintains clear logical structure, elucidating complex physical processes through detailed mathematical formulations and intuitive architectural diagrams. It appropriately situates the work within computational photography and defogging research contexts, ensuring strong readability. Regarding significance, this study addresses a critical challenge in preserving polarization information under scattering conditions, offering a novel technical pathway for applications demanding high physical consistency—such as autonomous driving and robotic vision—with broad practical potential. Regarding originality, the paper's contributions lie in redefining the physical model and introducing an innovative architecture that integrates differentiable physical modules into deep learning. This creative combination of existing techniques not only deepens our understanding of polarization imaging degradation mechanisms but also offers a novel perspective for joint image restoration tasks.

---

> ### Author Rebuttal · Authors · 2026-03-30
>
> We thank you for the insightful and constructive comments on our architectural choice, loss formulation, and physical robustness.
> > __Q1:__ Justification for using U-Net instead of a Transformer backbone.
>
> __A1:__ We thank the reviewer for highlighting this trade-off. Although atmospheric light estimation requires global context, our lightweight U-Net captures it efficiently through embedded self-attention (Sec. 4.2), without the high cost of full Transformers. To verify this, we replaced AL-Net's U-Net with a Swin-T under identical settings. As shown below, the Swin variant doubles parameters, halves inference speed, and degrades restoration metrics. We attribute this to the limited scale of high-quality paired polarimetric datasets. Under this data case, CNNs provide the crucial inductive biases (e.g., spatial locality) needed to prevent overfitting. That said, with much larger training data, Transformers may offer further gains in restoration and generalization. Overall, our hybrid U-Net provides the best balance between accuracy and efficiency for the current task. We will add this ablation and discussion to the supplementary material.
> |Setting|Params(M)|FPS|$S_0$(P/S)|DoLP(P/S)|AoLP(P/S)|
> -|-|-|-|-|-
> |Ours|12.2|16.8|33.70/0.948|28.80/0.762|19.43/0.501|
> |Ours+Swin|29.9|7.0|31.701/0.931|27.56/0.709|18.49/0.443|
> > __Q2:__ Clarification on the mathematical expansion of the complex phasor loss in Equation 22.
>
> __A2:__ We thank the reviewer for pointing out this missing detail. The complex phasor loss is computed by separating the real and imaginary parts, applying $\ell_1$ and $\ell_2$ norms independently, and then summing them, rather than using the complex modulus. Specifically, for a complex phasor $x \in \{C_A, C_D\}$ and its prediction $\hat{x}$, the composite loss $\mathcal{L}(x)$ expands as follows:$$\mathcal{L}(x) = \left( \|\text{Re}(x - \hat{x})\|_1 + \|\text{Im}(x - \hat{x})\|_1 \right) + 2 \left( \|\text{Re}(x - \hat{x})\|_2^2 + \|\text{Im}(x - \hat{x})\|_2^2 \right)$$ We will explicitly include this mathematical expansion below Equation 22 in the revised manuscript to guarantee reproducibility and remove any ambiguity.
> > __Q3:__ Separating atmospheric vs. intrinsic polarization for highly reflective objects.
>
> __A3:__ We thank the reviewer for this important question. Reflective materials are exactly one of the scenarios that motivated our generalized model design.
> + Our method models the signal as $I \cdot C_I = A \cdot C_A + D \cdot C_D$ , where the PRM recovers intrinsic polarization $C_D$ using airlight $C_A$ and transmission $K$. For reflective materials, the scene and airlight AoLP often mismatch ($\theta_A \neq \theta_D$). Traditional models assuming $\theta_A = \theta_D$ suffer from depolarization bias here (Appendix C.1). Our phasor model corrects this, stopping the network from mistaking local glares for global haze.
> + Our results already prove this robustness. In Figs. 5 and 8, the vehicles are mostly metal and glass. DuRP greatly boosts detection accuracy, showing it keeps the true polarization of objects rather than treating it as haze. To further support this, as detailed in our response to Reviewer L28b, DuRP significantly improves vehicle detection accuracy in real-world hazy conditions. Additionally, Fig. 7 shows our method successfully handles reflective items (e.g., glossy objects, glass buildings) without the color shifts or halos common in baseline models.
>
> We agree a dedicated test on highly reflective materials (glass, metal, water) will strengthen the paper. We will add these scenes to the revised appendix to directly prove our method's stability under strong local polarization.
> > __Q4:__ Robustness against complex sensor noise (e.g., spatial cross-talk).
>
> __A4:__ We thank you for the constructive feedback. To bridge the real-to-sim gap of the ideal Gaussian noise, we tested DuRP against realistic Poisson-Gaussian noise (shot+readout) and DoFP spatial cross-talk (leakage). For a stress test, we used severe parameters far exceeding standard 12-bit cameras ($0.005\le\sigma_g\le 0.015$) and consumer limits ($0.02\le\kappa\le0.10$). As shown below, DuRP remains highly stable. Under typical cross-talk ($\kappa=0.05$), DoLP PSNR drops just 1.17 dB (28.80 $\rightarrow$ 27.63). Even in the worst joint case ($\kappa=0.05, \sigma_g=0.015$), $S_0$ stays above 29 dB without collapsing. This robustness stems from our dual-stage physics design. Enforcing optical rules inherently resists local sensor artifacts better than black-box models. We will add these results to the appendix.
> |Setup|$\sigma_g$|$\kappa$|$S_0$(P)|DoLP(P)|AoLP(P)|
> -|-|-|-|-|-
> |Base|0|0|33.70|28.80|19.43|
> |P-G|0.005|0|31.41|26.04|17.47|
> |P-G|0.010|0|30.52|24.10|16.53|
> |P-G|0.015|0|29.27|22.07|15.78|
> |C-T|0|0.02|32.73|28.63|19.12|
> |C-T|0|0.05|32.23|27.63|19.00|
> |C-T|0|0.10|30.77|25.47|18.23|
> |Comb|0.005|0.05|30.97|26.04|17.39|
> |Comb|0.010|0.05|30.07|24.84|16.51|
> |Comb|0.015|0.05|29.20|23.50|15.81|

---

> > ### Author Rebuttal · Reviewer_XBwd · 2026-04-02
> >
> > I believe the author has thoroughly addressed all the issues I raised, providing clear explanations and relevant experimental data.

---

> > > ### Author Response · Authors · 2026-04-02
> > >
> > > Thank you again for your time and constructive feedback. We are glad that our responses have addressed most of your concerns.

---

### Official Review · Reviewer_KSju · 2026-03-13

**Soundness:** 2
**Presentation:** 3
**Significance:** 2
**Originality:** 3
**Overall Recommendation:** 4
**Confidence:** 3

**Summary:**

This paper studies polarization restoration in hazy environments and questions can polarization information be recovered jointly with radiance when atmospheric scattering corrupts both signals. This work proposes called DuRP, which derives a generalized polarization scattering model and embeds this model into a neural network. Specifically, the network estimates physical parameters and then inserts them into analytic reconstruction formulas to compute polarization and radiance. Experiments show higher PSNR and SSIM for the proposed method.

**Compliance With Llm Reviewing Policy:**

Affirmed.

**Final Justification:**

I would like to raise my score to 4 given the authors' feedback during the rebuttal. Most of my concerns have been solved.

**Key Questions For Authors:**

* The SRPG dataset is generated using the proposed generalized polarization model. Could this introduce bias toward the proposed method?

* How does the method perform on real polarized hazy images that are not generated by the proposed simulation pipeline?

* If the network directly predicts DoLP and AoLP using a standard architecture, how much performance is lost compared with the physics-guided formulation?

**Limitations:**

Yes

**Strengths And Weaknesses:**

Strengths

* The paper studies an valid and valuable problem. Polarization information is useful for tasks such as detection and shape estimation. Restoring polarization under haze can be relevant for many applications.

* The proposed method does not directly predict polarization but predicts parameters of the scattering model and then reconstructs the signals through analytic formulas, which is a valid design choice.


Weaknesses

* In this work, the network predicts physical parameters and the analytic equations are then applied as differentiable operators. Such designs seems to be common in physics-guided reconstruction problems. Thus the novelty seems limited to the specific polarization formulation.

* This work claims that previous models assume identical AoLP for airlight and scene light. However,  it is not clear how often this assumption fails in practice. More empirical analysis can be provided.

* It seems that the evaluation relies heavily on synthetic data. To my understanding, the dataset is generated using the proposed physics model, which raises a risk of model bias. A model derived from the same physics formulation may naturally perform better on such synthetic data.

* The dataset is relatively small. It seems that the SRPG dataset contains a limited number of scenes and synthetic samples. It is unclear whether the model generalizes to diverse real-world polarization conditions.

---

> ### Author Rebuttal · Authors · 2026-03-30
>
> We sincerely thank the reviewer for the thoughtful comments and constructive questions on our formulation, dataset bias, and real-world generalization.
> > __W1:__ Clarification of the main novelty.
>
> __A1:__ We agree that predicting physical parameters and reconstructing via differentiable operators is a common paradigm, but our novelty goes beyond merely swapping in a specific formulation. Existing methods usually assume that scene and airlight share the same AoLP, while we remove this limit and derive a generalized model. Beyond this, we map the coupled DoLP/AoLP into a complex phasor domain, which enables stable learning. In addition, we design a more physically grounded synthesis pipeline and build the SRPG dataset, which captures real-world polarimetric degradation more faithfully than simplified datasets. Thus, the contribution is not only the generalized polarization formulation, but also the phasor-based parameterization and the robust data pipeline that make joint restoration trainable and effective in practice.
> > __W2:__ How often does the assumption $\theta_A = \theta_D$ fail in practice?
>
> __A2:__ Thank you for the question. To assess how often the conventional assumption $\theta_A=\theta_D$ holds in practice, we performed an empirical analysis on 175 real-world scenes comprising over 137M valid pixels, and found that the strict equality assumption holds in only 14.82% of pixels, while most pixels have a clear angular mismatch. This is also expected physically: $\theta_D$ is mainly affected by surface geometry and material properties, whereas $\theta_A$ is mainly determined by scattering. Since these factors are generally different, exact alignment is generally rare. This supports relaxing the $\theta_A=\theta_D$ assumption in our generalized polarization model.
> |$\Delta\theta$|Proportion|
> -|-
> |< 1°|14.82%|
> |1°-10°|50.63%|
> |10°-20°|10.95%|
> |$\ge$ 20°|23.59%|
> > __W3\&Q1:__ Possible bias from SRPG generation.
>
> __A3:__ Thank you for this important concern. We agree that using a related physics model for synthesis can introduce evaluation bias, but believe this risk is limited here.
> * DuRP still solves the corresponding ill-posed inverse problem from degraded observations. This does not provide a trivial shortcut: recovering radiance and polarization remains highly underdetermined. In addition, all methods are trained and tested under the same data protocol, so DuRP’s advantage is not due to a different training setting.
> * DuRP generalizes beyond the synthetic distribution. We further evaluate it on real hazy scenes (e.g., Appendix Fig. 7), where DuRP still produces cleaner and more physically plausible results than prior methods. Consistently, our real-image no-reference evaluation provided in W4 also supports that the method does not rely solely on our synthetic data.
>
> Overall, although simulation bias is a valid concern in principle, our results suggest that DuRP is learning transferable restoration priors rather than merely overfitting to our synthetic data.
> > __W4\&Q2:__ Dataset scale and real-world generalization.
>
> __A4:__ Thank you for this comment. We agree that dataset scale matters for generalization. Although SRPG is smaller than standard RGB benchmarks, paired hazy/clean multi-polarization data are much harder to collect, and SRPG still provides meaningful supervision (1,290 pairs from 387 scenes). More importantly, generalization should be assessed on unseen real-world data. As shown in Fig. 7, DuRP produces stable and artifact-free resultst on real hazy scenes, suggesting that it is not limited to the synthetic conditions of SRPG. We further report no-reference image quality metrics on these real hazy images. DuRP achieves the best average rank, providing additional evidence that the model generalizes beyond the synthetic training set.
> |Method|NIQE↓|BRISQUE↓|MANIQA↑|DBCNN↑|Avg. Rank↓|
> -|-|-|-|-|-
> |SLP|5.62|24.27|0.325|0.461|7.5|
> |PLC|5.48|24.06|0.327|0.479|6.0|
> |LFP|5.73|21.45|0.345|0.505|4.5|
> |PD-Net|4.23|17.71|0.341|0.527|2.3|
> |Dehamer|5.15|25.62|0.340|0.491|5.3|
> |Deformer|5.71|22.87|0.330|0.475|6.5|
> |Selfpromer|5.39|26.29|0.365|0.534|4.0|
> |SGDN|5.72|25.27|0.338|0.479|6.8|
> |Ours|4.50|21.68|0.348|0.547|2.0|
> > __Q1:__ See W3\&Q1 above.
>
> > __Q2:__ See W4\&Q2 above.
>
> > __Q3:__ Performance loss when directly predicting DoLP and AoLP.
>
> __A3:__ Thank you for this key question. Removing all physical models and directly predicting target signals causes DoLP PSNR to drop by 4.44 dB and AoLP PSNR by 3.19 dB. A finer ablation that removes only the P-stage polarization model shows a similar trend: DoLP PSNR drops by 3.16 dB and AoLP PSNR by 2.65 dB. These results show that direct prediction fails to preserve the coupled polarization structure, confirming the necessity of our physics-guided design.
> |Setting|$S_0$(P/S)|DoLP(P/S)|AoLP(P/S)|
> -|-|-|-|
> |Ours|33.70/0.948|28.80/0.762|19.43/0.501|
> |w/o polar. physics|32.01/0.933|25.64/0.610|16.78/0.311|
> |w/o all physics|31.67/0.928|24.36/0.582|16.24/0.355|

---

> > ### Author Rebuttal · Reviewer_KSju · 2026-04-04
> >
> > Thanks for authors' effort. Yet it seems the rebuttal mainly highlights the advantage/strengths of the proposed methods, while missing in-depth discussions on the potential limitations or concerns, making it hard to fairly evaluate the proposed method.
> >
> > For example,  the rebuttal acknowledges that ``We agree that using a related physics model for synthesis can introduce evaluation bias''. I'm wondering what types of the data is adopted in the compared methods, do they also (potentially) suffer from the risk of model bias? The results can be biased with the proposed setting, even if it is applied over all the compared methods.
> >
> > The rebuttal also acknowledges that ``the dataset scale matters for generalization'' and  SRPG is kind of small. While it is hard to solve such a concern in a short rebuttal period.

---

> > > ### Author Response · Authors · 2026-04-04
> > >
> > > Thank you again for your time and constructive feedback.
> > > > __Q1.1:__ What types of data are adopted by the compared methods?
> > >
> > > __A1.1:__ The compared methods use different input forms, but all are evaluated on the same SRPG scenes and haze synthesis. Specifically:
> > > * Traditional methods (SLP, LFP): single-image methods with one RGB input. We apply them to the hazy RGB image at each of the four polarization angles separately, then compute DoLP and AoLP from the four restored outputs.
> > > * Learning-based single-image methods (Dehamer, Deformer, Selfpromer, SGDN): single-image RGB dehazing networks. They are retrained on SRPG-derived RGB data, with each polarization angle treated as a separate sample rather than a joint four-image input. During test, the four hazy RGB images are also processed separately, and DoLP/AoLP are derived from the four outputs.
> > > * Polarization-based methods (PD-Net, PLC): polarization-based methods that take four-angle polarization images as input. PD-Net is retrained and tested on SRPG-derived polarization inputs. PLC is training-free and is tested directly on the SRPG test set.
> > >
> > > Therefore, all methods are compared on the same scenes and haze process, while keeping their original input setting.
> > > >__Q1.2:__ Do the compared methods also face model bias risk?
> > >
> > > __A1.2:__ Thank you for raising this important point. Compared methods in our paper are subject to the same underlying physical model — Atmospheric Scattering Model. Specifically:
> > > * Traditional methods (SLP, LFP) explicitly invert Atmospheric Scattering Model during test. Their core algorithmic assumption is this model, so they are at least as tightly coupled to it as DuRP.
> > > * Learning-based single-image methods (Dehamer, Deformer, etc.), in their original publications, were trained on synthetic datasets (e.g., RESIDE) that are also generated under the atmospheric scattering model. In our setting, they are retrained on SRPG, which is derived from the same physical family.
> > > * Polarization-based methods (PD-Net, PLC) also build upon atmospheric scattering, extended with polarization terms.
> > >
> > > Therefore, this risk should be viewed as a shared issue rather than a phenomenon that benefits only DuRP, since all compared methods are tied, to some extent, to the atmospheric scattering model. We also agree that using the same benchmark does not eliminate this bias; it only ensures a relatively fair comparison across methods.
> > > >__Q1.3:__ The results can be biased with the proposed setting, even if it is applied over all the compared methods.
> > >
> > > __A1.3:__ Using the same setting for all methods makes the comparison fairer, but it cannot fully remove bias. While the compared methods, our method and our SRPG dataset rely on the atmospheric scattering model, varying degrees of unquantifiable bias may still exist. We therefore treat the synthetic results only as evidence under this controlled setting, rather than as full proof of performance in all real-world conditions. The real-scene results are included only as supporting evidence of transfer, not as conclusive validation, because clear-scene polarization ground truth is unavailable in uncontrolled hazy scenes. In the revision, we will more explicitly articulate this scope and the potential limitations to avoid overstating our generalization capabilities.
> > > >__Q1.4:__ SRPG is kind of small, and it is hard to solve such a concern in a short rebuttal period.
> > >
> > > __A1.4:__ We agree that SRPG is relatively small. We therefore do not claim that the current benchmark fully covers real-world polarization conditions. In the revision, we will make this limitation clearer. At the same time, we would like to highlight that this limitation primarily pertains to the current dataset scale rather than the synthesis pipeline itself. Our physically-grounded pipeline is inherently scalable and can be readily extended as more clear polarization data becomes available to the community.

---

### Official Review · Reviewer_RgPg · 2026-03-13

**Soundness:** 3
**Presentation:** 3
**Significance:** 2
**Originality:** 3
**Overall Recommendation:** 4
**Confidence:** 4

**Summary:**

This paper introduces DuRP, a two-stage deep learning framework for recovering scene radiance and polarization from hazy images. It tackles the flawed AoLP invariance assumption in prior work by embedding newly derived, generalized physical models as differentiable network layers. Experiments show DuRP achieves state-of-the-art results in polarimetric restoration, significantly benefiting downstream tasks like 3D shape estimation. A new, realistic dataset, SRPG, is also contributed.

**Compliance With Llm Reviewing Policy:**

Affirmed.

**Final Justification:**

After carefully considering both the original submission and the response, I will raise my score to 4.

**Key Questions For Authors:**

1. How does DuRP perform on polarimetric images captured by non-DoFP polarimetric cameras (e.g., DoA, DoT)?
2. The SRPG dataset is synthesized with DepthAnything for depth estimation—how sensitive is DuRP’s performance to the accuracy of the depth map used in the synthesis pipeline? Would lower-quality depth estimates (e.g., from monocular depth models with higher error) significantly degrade the model’s generalization to real-world data?
3. For real-world applications (e.g., autonomous driving), DuRP requires real-time inference. What optimizations (e.g., model quantization, pruning, lightweight backbone replacement) have been tested to improve inference speed, and what is the minimal hardware requirement for real-time performance?

**Limitations:**

yes

**Strengths And Weaknesses:**

Strengths:
1. The paper re-derives the polarimetric scattering imaging process and formulates generalized models that relax the unrealistic AoLP invariance assumption of traditional methods. By introducing a complex phasor representation for polarimetric states, the non-linear coupling between DoLP and AoLP is linearized, providing a mathematically rigorous and physically plausible foundation for joint restoration
2. The DuRP framework decomposes the inverse restoration problem into sequential Polarization Reconstruction (P-Stage) and Radiance Reconstruction (R-Stage), with differentiable physical model operators embedded as inductive biases.
3. The authors construct the SRPG dataset via a physics-grounded synthesis pipeline that simulates complex real-world polarimetric distributions, addressing the deficiency of existing polarized haze datasets that use oversimplified pseudo-polarization labels.

Weaknesses:
1. The paper lacks a discussion on the computational cost, inference time, or model size compared to simpler baselines. This complexity might hinder its adoption in resource-constrained or real-time applications.
2. The proposed synthesis pipeline for the SRPG dataset relies on "normalized scene depth" to compute the transmission map. In the real-world application, the method itself does not estimate depth. The paper does not discuss how the method might perform if the training data's depth statistics differ significantly from those in a real-world test scene.
3. The paper acknowledges performance degradation in dense fog (extreme haze) where polarization signals approach zero, but provides no detailed experimental analysis.
4. The framework is designed for Division of Focal Plane (DoFP) polarimetric cameras, but there is no exploration of its generalization to other polarimetric camera types (e.g., Division of Amplitude, Division of Time). This limits the applicability of DuRP to different polarimetric sensing setups.

---

> ### Author Rebuttal · Authors · 2026-03-30
>
> We sincerely thank the reviewer for recognizing our core contributions and providing constructive feedback to strengthen this paper.
> > __W1:__ Computational cost and inference time.
>
> __A1:__ Thank you for this comment. We benchmark DuRP against strong baselines on the SRPG test set using an RTX 4090. Although DuRP is not the fastest, it gives a good speed-quality trade-off: it is smaller than several baselines and much faster than Dehamer, SGDN, and Selfpromer, while achieving better joint radiance-polarization restoration.
> ||PD-Net|Dehamer|Deformer|Selfpromer|SGDN|Our|
> |-|-|-|-|-|-|-|
> |Params(M)|44.7|29.4|4.6|34.0|13.3|12.2|
> |Runtime(ms)|23.9|78.4|43.3|2410|144.8|59.6|
> > __W2 & Q2__: Reliance on synthesized depth and sensitivity to depth quality.
>
> __A2:__ Thank you for this comment. We agree that the effect of depth quality in synthesis should be clarified. First, in SRPG, depth is used only in synthesis to form the transmission map $t(x)=\exp(-\beta z(x))$. Since $\beta$ is sampled from a wide range, the synthesis mainly depends on relative depth, rather than absolute depth. Second, DuRP does not estimate or use depth at inference time. This makes DuRP less dependent on training depth statistics. To further verify this, we kept the model fixed and perturbed the depth maps only when building the test set, i.e., we re-generated hazy test images from perturbed depth and evaluated the same model without retraining. We considered three perturbation types: smooth, affine, and low-freq. As the results indicate, the performance drop remains moderate even under large perturbations. This suggests that DuRP is not tied to a specific depth distribution and does not rely on highly accurate depth to generalize reasonably well.
> |Perturb|$\sigma$|$S_0$(P)|DoLP(P)|AoLP(P)|
> |-|-|-|-|-|
> |none|0.00|33.70|28.80|19.43|
> |smooth|0.10|32.51|28.35|18.65|
> |smooth|0.20|31.60|27.80|18.34|
> |affine|0.10|33.09|28.13|18.63|
> |affine|0.20|32.41|27.65|18.43|
> |low-freq|0.10|32.20|28.04|18.50|
> |low-freq|0.20|30.82| 27.25|18.19|
> > __W3:__ Performance degradation in dense fog.
>
> __A3__: Thank you for pointing out this important edge case. We agree that it needs clearer analysis. The degradation in very dense fog is mainly a physical limit, not a failure unique to DuRP. As haze becomes very dense $(t \rightarrow 0)$, scene light is heavily attenuated and scene polarization approaches zero. In this case, polarimetric cue becomes unreliable. Thus, this is a fundamental challenge for all polarization-based methods, not only ours. To make this clear, we will add a failure-case analysis in the revision, including dense-fog examples and DoLP histograms to show how performance degrades as measurable polarization vanishes.
> >__W4 & Q1:__ Generalization to other polarimetric camera types (DoA/DoT).
>
> **A4:** Thank you for raising this point. We agree that this should be clarified more explicitly. From the theory side, DuRP is not tied to a specific polarimetric camera type. It takes four polarized images $(I_0,I_{45},I_{90},I_{135})$ as input, which can in principle be provided by DoFP, DoA, or DoT systems. Thus, DuRP is sensor-agnostic at the measurement level. The gap in our extra experiment mainly comes from DoFP sampling and demosaicing, not from the DuRP formulation itself. Our test images are only $512\times512$, so converting full-resolution four-angle data into simulated DoFP data greatly reduces the information in each polarization view (roughly to $128\times128$ for our $512\times512$ test images). The following EARI demosaicing then adds blur and artifacts under such sparse sampling. Hence, the gap is mainly caused by demosaicing loss before DuRP, while DuRP itself remains theoretically applicable to DoFP/DoA/DoT cameras.
> ||$S_0$(P/S)|DoLP(P/S)|AoLP(P/S)|
> |-|-|-|-|
> |DoT/DoA(Full-Res,Original)|33.70/0.948|28.80/0.762|19.43/0.501|
> |DoFP(Mosaicked+Demosaiced)|28.52/0.911|16.90/0.493|13.98/0.334|
> > __Q1:__ See W4\&Q1 above.
>
> > __Q2:__ See W2\&Q2 above.
>
> > __Q3:__ Real-time inference, tested optimizations, and minimal hardware requirements.
>
> __A3:__ Thank you for this practical question. We agree that real-world deployment needs low-latency inference, so we conducted a preliminary acceleration study. Among the tested settings, FP16 (AMP) is the best practical option, raising speed from 16.8 FPS to 26.4 FPS with negligible accuracy change. By contrast, FP16 + `torch.compile` is faster but shows a clear drop in quality in our current setup. For hardware, we only tested DuRP on one RTX 4090, which achieves 26.4 FPS with AMP at baseline-level accuracy. Thus, this is the minimal validated hardware in our current study, not a strict lower bound. Achieving stable 30+ FPS will likely require further model-side optimization.
> ||ms|FPS|$S_0$ (P/S)|DoLP (P/S)|AoLP (P/S)|
> |-|-|-|-|-|-|
> |FP32(Base)|59.6|16.8|33.70/0.948|28.80/0.762|19.43/0.501|
> |FP16(AMP)|37.8|26.4|33.69/0.948|28.80/0.762|19.43/0.501|
> |FP16+Compile|19.0|52.6|24.34/0.822|21.13/0.569|16.21/0.362|

---

> > ### Author Rebuttal · Reviewer_RgPg · 2026-04-05
> >
> > Thanks for the authors’ response. The rebuttal has addressed my questions.

---

> > > ### Author Response · Authors · 2026-04-07
> > >
> > > Thank you very much for the positive feedback. We are glad that our rebuttal has addressed your concerns.

---

### Decision · Program_Chairs · 2026-04-30

**Decision:**

Accept (regular)

**Comment:**

This paper studies polarization restoration in hazy environments. Which originated from 1999, before deep learning. The authors derive a more universal generalized polarization physical model and integrate it as an inductive bias into deep networks. The authors propose a synthetic dataset to validate their method.
This paper receives mixed reviews. Three weak accepts and one weak rejects. Reviewers recognise the strength that it is a valid problem, and the theoretical foundations in physics are provided. The method sounds, and the proposed synthetic datasets add value for future research. One reviewer maintains weak rejection given the concern that it lacks concrete validation on real data.
In the end, there is no consensus among reviewers. The AC has checked the paper, reviews, rebuttals. The AC finds the merit outweighs the demerit. The AC therefore recommends weak acceptance.